JCB Journal of Cell Biology

# Submembrane liprin-α1 clusters spatially localize insulin granule fusion

Kylie Deng[1], Kitty Sun[1], Nicole Hallahan[1], Wan Jun Gan[1], Michelle Cielesh[1], Baharak Mahyad[2], Melkam A. Kebede[1], Mark Larance[1], and Peter Thorn[1]

Insulin granule fusion in pancreatic β cells localizes to where they contact the ECM of the islet capillaries. The mechanism(s) underpinning localization are unclear. Using glucose or high K⁺ stimulation or the global uncaging of Ca²⁺, we show granule fusion consistently focused to the β cell–ECM interface, suggesting a specific localization mechanism. We tested for the involvement of liprin-α1, a scaffold protein enriched at the β cell–ECM interface. Liprin-α1 knockdown did not affect high K⁺-stimulated insulin secretion but did impair localization of exocytosis. Liprin-α1 knockdown impaired glucose-induced insulin secretion with evidence that the C-terminal of liprin-α1 positions liprin-α1 in clusters at the β cell–ECM interface. Liprin-α1 cluster size and number are regulated by glucose, and exocytosis is spatially coupled with the clusters. Immunoprecipitation and mass spectrometry characterized a liprin-α1 interactome, including β2-syntrophin, an insulin granule–linked protein. We conclude that liprin-α1 is part of a complex that is regulated by glucose and locally targets insulin granules to the β cell–ECM interface.

## Introduction

Insulin granule exocytosis in pancreatic β cells is a highly coordinated process requiring the integration of a myriad of inputs to regulate secretory output (Rorsman and Ashcroft, 2018). Like all endocrine systems, islets of Langerhans are richly vascularized and possess a dense capillary network that is intimately associated with β cells (Dolenšek et al., 2015; Lammert and Thorn, 2020). This region where β cells contact ECM proteins of the islet capillary network (herein termed ECM interface) constitutes ∼15% of the total β cell membrane area, and accumulating evidence demonstrates that insulin granule fusion is focused to this region (Jevon et al., 2022; Low et al., 2014), thereby delivering insulin directly into the blood stream. The mechanisms that localize insulin granule fusion to the ECM interface are unknown.

Whether insulin granule fusion is localized to the capillary interface has been debated over the years (Bonner-Weir, 1988; Jevon et al., 2022; Low et al., 2014; Rutter et al., 2006; Takahashi et al., 2002). Early work showed an enrichment of granules at the β cell/capillary interface following chronic stimulation, suggesting polarized secretion toward this region (Bonner-Weir, 1988). This was later disputed by a study using live-cell two-photon microscopy demonstrating preferential granule fusion away from the capillaries (Takahashi et al., 2002), supported by another study using 3D confocal microscopy in MIN6 cells (Rutter et al., 2006). However, limitations of these studies include the use of 2D imaging or isolated cells, both of which do not account for the complex 3D relationship β cells have with capillaries within the islet environment. New approaches to study β cells in situ using pancreatic slices and in culture models that mimic the ECM interface now provide very strong evidence that granule fusion is specifically enriched at the capillary ECM interface of β cells (Gan et al., 2018; Jevon et al., 2022).

The mechanisms underlying this local enrichment of granule fusion remain to be discovered. One possibility is that the molecular machinery of granule fusion and/or Ca²⁺ pathways might be localized. The essential molecular machinery of insulin granule fusion bears a strong resemblance to that of synaptic vesicle release, including SNARE complex proteins, vesicular proteins Rab3/27, and voltage-gated Ca²⁺ channels (VGCCs) (Lang, 2001). A rise in Ca²⁺ by entry through VGCCs is the main trigger for insulin granule fusion (Schulla et al., 2003). Direct measurements show that both syntaxin 1A, the primary plasma membrane SNARE (Nagamatsu et al., 1996), and Ca$_V$1.2, the primary calcium channel subtype in mouse β cells (Rorsman et al., 2012), are distributed uniformly across the β cell membrane (Low et al., 2014; Ohara-Imaizumi et al., 2019). And, therefore, although SNAREs, like syntaxin 1A, might very locally cluster at sites of exocytosis (Gandasi and Barg, 2014), their wide distribution suggests that regional enrichment does not underpin localized targeting of exocytosis to the ECM interface.

[1]School of Medical Sciences, Charles Perkins Centre, University of Sydney, Camperdown, Australia; [2]Nikon Australia Pty Ltd, St Kilda, Australia.

Correspondence to Peter Thorn: peter.thorn@sydney.edu.au; Kylie Deng: kylie.deng@svi.edu.au.

If the molecular machinery of granule fusion and Ca²⁺ pathways cannot explain the regional enhancement of granule fusion, it might involve either the local delivery of granules to this region or the local regulation of Ca²⁺ channel activity, or both. In neurones, synaptic vesicle release is tightly restricted in the presynaptic active zone (AZ) by an evolutionarily conserved protein complex, including scaffold proteins liprin-α1, ELKS, RIM2 and piccolo (Südhof, 2012). This presynaptic complex both tethers synaptic vesicles prior to granule docking and recruits Ca²⁺ channels to locally deliver Ca²⁺ as the trigger for vesicle exocytosis (Südhof, 2012). The localization of VGCCs to vesicle release sites generates local nanodomains of high [Ca²⁺] that permit rapid exocytosis (Eggermann et al., 2011; Südhof, 2012). Whether analogous mechanisms exist in β cells is unknown.

Work to date supports the idea that presynaptic-like mechanisms might exist in β cells. In both mouse and human β cells, the ECM interface is a region enriched in presynaptic scaffold proteins, including liprin-α1, ELKS, RIM2, and piccolo (Cottle et al., 2021; Low et al., 2014; Ohara-Imaizumi et al., 2005). Studies show that mouse β cell–specific knockout of RIM2 (Yasuda et al., 2010) or ELKS (Ohara-Imaizumi et al., 2019) impairs glucose-dependent insulin secretion. There also appears to be a close association between Ca$_V$1.2 and insulin secretory granules (Bokvist et al., 1995; Gandasi et al., 2017). Furthermore, *in vitro*–binding studies in MIN6 cells indicate the formation of a ternary complex with ELKS, RIM2, and bassoon (Ohara-Imaizumi et al., 2005) and a direct interaction between RIM2 and Ca$_V$1.2 (Shibasaki et al., 2004). There is also evidence for an interaction between ELKS and the auxiliary β subunit of VGCCs, and that this ELKS–VGCC interaction regulates Ca²⁺ channel function and is required for local Ca²⁺ influx at the β cell–ECM interface (Jevon et al., 2022; Ohara-Imaizumi et al., 2019). This mechanism to locally regulate Ca²⁺ channels in β cells is consistent with one of the key functions of the presynaptic scaffold complex and alone may be sufficient to enable the local enhancement of granule fusion in this region. However, in neurones, the presynaptic complex also acts to position and tether vesicles, and the question therefore arises as to whether a similar mechanism to position granules at the ECM interface exists in β cells.

Here, we test the hypothesis that a presynaptic-like scaffold protein complex leads to the focus of insulin granule fusion to the β cell–ECM interface. Using glucose and high K⁺ stimulation as well as the global liberation of caged Ca²⁺, we show that insulin granule fusion is consistently localized to the β cell–ECM interface, directly demonstrating that a mechanism of granule positioning exists. Knockdown of liprin-α1, a key component of the presynaptic complex, in β cells impairs the localization of granule fusion and reduces both phases of glucose-stimulated insulin secretion. Moreover, we show the C terminus of liprin-α1 is essential for positioning liprin-α1 to the β cell–ECM interface, where it assembles in dynamic glucose-dependent clusters that spatially constrain granule fusion. With a candidate approach, co-immunoprecipitation (IP) of liprin-α1 followed by mass spectrometry analysis identified protein–protein interactions indicative of a broader complex suggestive of presynaptic-like control and potential coupling to insulin granules *via* an interaction with a granule-linked protein β2-syntrophin. Together, we conclude that liprin-α1 regulates insulin secretion in pancreatic β cells and propose it acts to localize insulin granule fusion to the β cell–ECM interface.

## Results

### Evidence for a distinct mechanism that localizes insulin granule fusion to the β cell–ECM interface

Within islets, the ECM is present as a basement membrane that is enriched around the capillaries and not present between adjacent endocrine cells (Nikolova et al., 2006). As such, the β cell–capillary interface is the only region of the cell that contacts ECM and is the exclusive site of integrin activation (Gan et al., 2018). We have previously shown that the local activation of integrins is necessary for the localization of both insulin granule fusion and presynaptic scaffold proteins like liprin-α1 and ELKS to the ECM interface (Gan et al., 2018; Jevon et al., 2022). Both local granule fusion and positioning of presynaptic scaffold proteins in β cells can be observed in pancreatic slices (Jevon et al., 2022). But, as a system more amenable to manipulation, we have used a culture of isolated β cells on an ECM-coated surface as a model system and have shown this recapitulates localized granule fusion and enrichment of presynaptic scaffold proteins to the β cell–ECM interface (Gan et al., 2018; Jevon et al., 2022).

Our study uses isolated mouse islets that are further broken down to single islet cells, the majority of which are β cells (see Materials and methods). Here, we replicated the findings that culturing of these cells on laminin-511 shows enrichment of liprin and ELKS at the β cell–ECM interface (Fig. 1 A). We counterstained for insulin to positively identify β cells, and, as expected because of the huge abundance of insulin granules, we observe insulin staining throughout the cells (Fig. 1 A). We then used 3D live-cell two-photon microscopy to identify each insulin granule exocytotic event in time and space by tracking the entry of extracellular fluorescent dye sulforhodamine B (SRB) into each fusing granule (Fig. 1 B) (Low et al., 2014). Continuous recording over 15 min following high glucose (16.7 mM) or high K⁺ (40 mM) stimulation led to the identification of exocytotic events. When mapped in space, we observed a significant bias of granule fusion events toward the β cell–ECM interface (Fig. 1, C and D), as shown in comparisons of the exocytotic density at the β cell–ECM interface (bottom) compared with rest of the cell (upper, Fig. 1 G). We conclude that β cells orientate with respect to the ECM interface, and both position presynaptic scaffold proteins and localize insulin granule fusion to this interface.

Polarized Ca²⁺ influx (Ohara-Imaizumi et al., 2019) and fast intracellular Ca²⁺ waves (Jevon et al., 2022) originate at the β cell/capillary interface and likely reflect a local clustering of active Ca²⁺ channels (Jevon et al., 2022). To test whether localized Ca²⁺ entry could drive the localized granule fusion we observe, we sought to identify the sites of insulin granule exocytosis independent of Ca²⁺ entry by stimulating the cells with the global photolytic release of caged Ca²⁺ (Ca²⁺-NP-EGTA) (Ellis-Davies and Kaplan, 1994). β cells were cultured onto laminin-511–coated coverslips and loaded with NP-EGTA, and we used 3D two-photon microscopy to map the sites of β cell

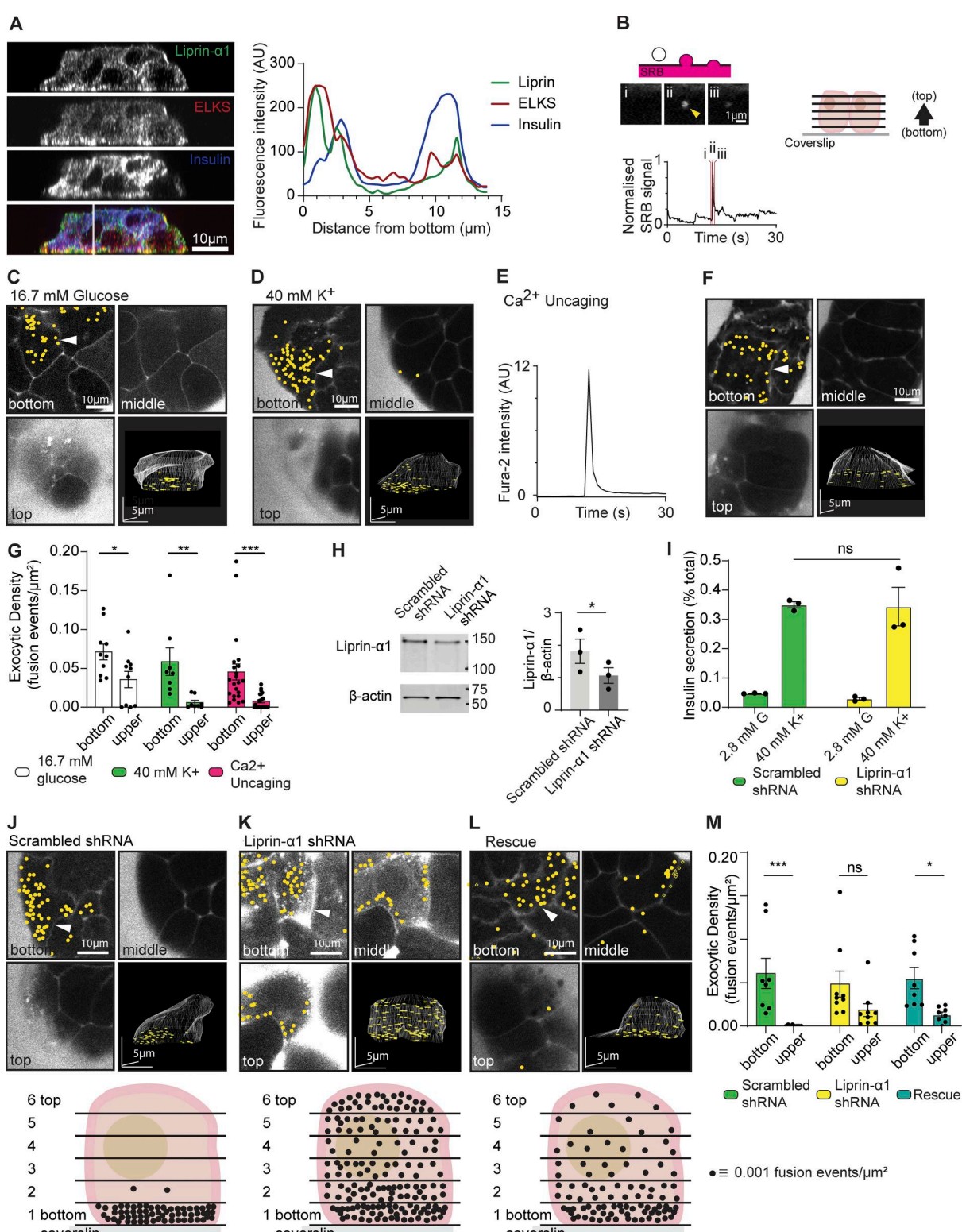

Figure 1. **Liprin-α1 knockdown in β cells impairs localization of granule fusion. (A)** Immunofluorescence staining of presynaptic scaffold proteins liprin-α1 (green) and ELKS (red) in isolated mouse β cells (insulin; blue) grown on coverslips coated with ECM (laminin-511). A line scan plotting fluorescence intensity across an orthogonal section (XZ) shows local enrichment of liprin-α1 and ELKS, but not insulin, at the laminin-cell interface. **(B)** When stimulated, isolated β cells bathed in an extracellular dye (SRB) and imaged with two-photon microscopy show short-lasting flashes of fluorescence as individual granules fuse with the membrane and SRB enters each fusing granule. 3D live-cell two-photon imaging with z stacks (2-μm apart) was used to record β cell exocytotic events in time and space, as shown in the cartoon. **(C)** Continuous recording over 15 min during stimulation with 16.7 mM glucose led to the identification of many exocytotic events, marked with yellow dots. An exemplar image of one cell shows three image planes (bottom, middle, and top) as well as a 3D projection image of the whole cell and highlights a strong bias of events at the ECM-cell interface (bottom). **(D)** Stimulation with 40 mM K⁺ for 15 min also induced many exocytic

events, again in the exemplar images showing a strong bias to the ECM-cell interface (bottom). **(E)** β cells were loaded with the photolabile $Ca^{2+}$ chelator nitrophenyl EGTA (NP-EGTA) and Fura-2, AM, enabling the UV flash photolysis-catalyzed global intracellular uncaging of $Ca^{2+}$. **(F)** $Ca^{2+}$ uncaging by a 100-ms UV flash triggered a rapid transient increase in intracellular $[Ca^{2+}]$, tracked with Fura-2, and (F) induced many exocytotic events, again with a bias of events to the ECM-cell interface (bottom). **(G)** Histogram of exocytotic density at the cell bottom versus upper planes (the average of planes 2–6) shows granule fusion is significantly biased toward the ECM-cell interface (bottom) for all three stimulation conditions ($n \geq 8$ β cells obtained from $\geq 3$ animals, two-way ANOVA followed by Bonferroni's multiple comparisons test). **(H)** Western blot showing liprin-α1 expression in mouse islets infected with adenovirus encoding GFP-scrambled shRNA (control) and GFP–liprin-α1 shRNA. Quantification of liprin-α1 protein expression normalized to β-actin is shown as a histogram ($n = 3$; paired two-tailed Student's $t$ test). **(I)** Control β cells and cells after liprin-α1 knockdown were stimulated with 40 mM $K^+$ for 30 min. Liprin-α1 knockdown had no effect on high $K^+$-induced insulin secretion, normalized to total cellular insulin content (for each condition, $n = 3$ animals; two-way ANOVA followed by Tukey's multiple comparisons test). **(J–M)** High $K^+$-induced granule fusion events were recorded using 3D live-cell two-photon microscopy. **(J)** An exemplar of a control cell (scrambled shRNA) showed a significant bias of granule fusion events at the ECM-cell interface (bottom) compared with upper regions. A schematic diagram summarizes the dataset ($n = 3$ animals, 8 cells) by showing the average distribution of granule fusion events at each image plane, each dot representing an exocytic fusion density of 0.001 events μm$^2$. **(K)** In contrast, liprin-α1 knockdown cells had a relatively even distribution of granule fusion events around the whole cell, as shown in the example images and in the schematic diagram ($n = 3$ animals, 9 cells). **(L)** Re-expression of human GFP–liprin-α1 after knockdown partially rescued granule targeting to the ECM-cell interface (bottom), as shown in the example images and in the schematic diagram ($n = 4$ animals, 11 cells). **(M)** A histogram of the data in J–L show significant differences in the high $K^+$-induced exocytic density between the upper sections and the ECM-cell interface (bottom) for control cells that was abolished with liprin-α1 shRNA and partially, but significantly, restored with the human liprin-α1 re-expression (for each condition, $n = 3–4$ animals; two-way ANOVA followed by Bonferroni's multiple comparisons test). All data are shown as mean ± SEM. Source data are available for this figure: SourceData F1.

granule fusion triggered by global uncaging of $Ca^{2+}$ by a 100-ms UV flash. In response to the UV flash, we observed a sharp rise in intracellular $[Ca^{2+}]$ (Fig. 1 E) followed by granule fusion events (Fig. 1 F), which, when mapped, showed a significant bias in frequency toward the β cell–ECM interface (Fig. 1 G).

### Liprin-α1 knockdown in β cells impairs the localization of insulin granule fusion

The above data identify that a mechanism(s) exists, independent of localized $Ca^{2+}$ entry, to localize insulin granule fusion to the β cell–ECM interface. In neurones, synaptic vesicle localization is accomplished by presynaptic scaffold protein complexes that tether synaptic vesicles prior to vesicle docking and fusion at the cell membrane (Südhof, 2012). Among the presynaptic scaffold proteins, liprin-α plays a central role in the organization of protein assemblies of the AZ (Emperador-Melero et al., 2021; Liang et al., 2021; Wei et al., 2011; Xie et al., 2021) and in anchoring synaptic vesicles in the presynaptic domain (Wong et al., 2018). To date, although liprin-α1 has been identified in β cells and is enriched at the β cell–ECM interface (e.g., Fig. 1 A), nothing is known about its function.

To identify if liprin-α1 plays a role in the localization of insulin granule fusion, we examined the effect of liprin-α1 knockdown. We infected β cells with either GFP-scrambled shRNA (control) or GFP–liprin-α1 shRNA adenovirus and show a ~42% knockdown in liprin-α1 expression compared with controls using western blot (Fig. 1 H). Knockdown of liprin-α1 had no effect on high $K^+$-induced insulin secretion (Fig. 1 I), suggesting that liprin-α1 in β cells, like in neurons, is not involved in the final stages of granule docking or exocytosis. To test for an effect on localization of granule fusion, we again used live-cell 3D two-photon microscopy. Mapping the sites of β cell granule fusion in 3D, in response to high $K^+$ stimulation, we show there is a significantly greater density of granule fusion events at the β cell–ECM interface in control cells infected with scrambled shRNA adenovirus (Fig. 1, J and M), but this localization is disrupted in cells infected with liprin-α1 shRNA (Fig. 1, K and M) and partially, but significantly, rescued when liprin-α1 is re-expressed (re-expression of human GFP–liprin-α1, Fig. 1,

L and M). We suggest that liprin-α1 regulates a mechanism that localizes granules to the β cell–ECM interface prior to docking but, like liprin-α in neurones, is not involved in the final stages of granule docking or fusion (Südhof, 2012).

### Liprin-α1 knockdown in β cells impairs glucose-induced insulin secretion

High $K^+$ stimulation in β cells triggers a transient burst of insulin secretion (<5 min) by depolarization and opening of VGCCs and primarily stimulates the fusion of a population of granules that are stably docked at the membrane (Shibasaki et al., 2007). Glucose stimulation also depolarizes β cells but uses additional mechanisms that regulate and augment insulin secretion over periods of time (>30 min) and induce the fusion of a population of mobile granules (Gaisano, 2017; Shibasaki et al., 2007). Given the distinct nature of the two stimuli, we wanted to test the impact of liprin-α1 knockdown on glucose-induced insulin secretion. Our results show that after liprin-α1 knockdown there was a significant reduction in insulin secretion in a static (30-min high-glucose stimulation) assay that was rescued by the re-expression of human GFP–liprin-α1 (Fig. 2 A). In a perifusion assay, measuring insulin secretion over time, this reduction of glucose-induced insulin secretion was observed for both first-phase insulin secretion (<10 min) and second-phase insulin secretion (Fig. 2, B–E). We conclude that liprin-α1 plays a specific role in glucose-dependent control of insulin secretion.

The frequency of granule fusion events in the liprin-α1 knockdown cells was too low when using glucose as a stimulus to use the 3D live-cell assay to map fusion events. However, we tested the possibility that liprin-α1 might affect other pathways. In neurons, presynaptic scaffold proteins not only position synaptic vesicles but also locally recruit $Ca^{2+}$ channels (Südhof, 2012). Therefore, we tested if liprin-α1 knockdown might affect β cell $Ca^{2+}$ responses. β cells were loaded with Fura-2 to measure intracellular $[Ca^{2+}]$ (Grynkiewicz et al., 1985), and in both control (GFP-scrambled shRNA) and liprin-α1 knockdown cells, we observed robust $Ca^{2+}$ responses following high-glucose stimulation (Fig. 2 F) with no differences in area under curve (Fig. 2 G), baseline or peak $[Ca^{2+}]$, and latency (time to peak) (Fig. S1).

Figure 2. **Liprin-α1 knockdown in mouse β cells reduces both phases of glucose-stimulated insulin secretion. (A)** Static incubation of β cells infected with adenovirus encoding GFP-scrambled shRNA (control) or GFP–liprin-α1 shRNA in 16.7 mM glucose for 30 min showed a significant reduction in insulin secretion, normalized to total cellular insulin content, after liprin-α1 knockdown. Re-expression of GFP-human–liprin-α1 rescued this secretory defect (for each condition, n = 3 animals; two-way ANOVA followed by Tukey's multiple comparisons test). **(B–E)** Dynamic glucose-stimulated insulin secretion profiles from perifused β cells. Insulin secretion in both first phase (10 min following stimulation) and second phase (20–40 min following stimulation) was reduced after liprin-α1 knockdown, quantified by measuring area under curve (AUC). **(F and G)** β cells were loaded with Fura-2 to measure intracellular [Ca²⁺]. In both liprin-α1 knockdown and control groups, robust Ca²⁺ responses were recorded following stimulation with 16.7 mM glucose with no difference in area under curve (n ≥ 9 β cell clusters from 3 animals; Student's t test, unpaired, equal variance.

These results indicate that, under the conditions tested, the primary action of liprin-α1 knockdown to reduce glucose-induced insulin secretion is not on the Ca²⁺ responses.

### The C terminus of liprin-α1 positions liprin-α1 to the ECM interface

All liprin-α isoforms share a similar domain organization, consisting of an N-terminal coiled-coil region and a highly conserved C-terminal region comprised of three sterile alpha motif domains (Serra-Pagès et al., 1998). This C-terminal region mediates interactions with membrane phosphatases (e.g., LAR, PTP

δ, and PTPσ) (Serra-Pagès et al., 1998) and, importantly, interacts with liprin-βs (Serra-Pagès et al., 1998) to locate liprin-α1 to focal adhesions (Bouchet et al., 2016; van der Vaart et al., 2013). To test the importance of the C-terminal to the targeting and function of liprin-α1 in β cells, we generated an N-terminal truncated mutant (liprin-N, aa 1–492) lacking sterile alpha motif domains (Fig. 3 A). Overexpression of GFP-liprin–full-length (FL) and GFP-liprin-N (N terminus only) in isolated mouse β cells was quantified using western blot (Fig. 3 B). When the β cells were cultured on laminin-511–coated coverslips, GFP-liprin-FL was enriched at the β cell–ECM interface (Fig. 3 C),

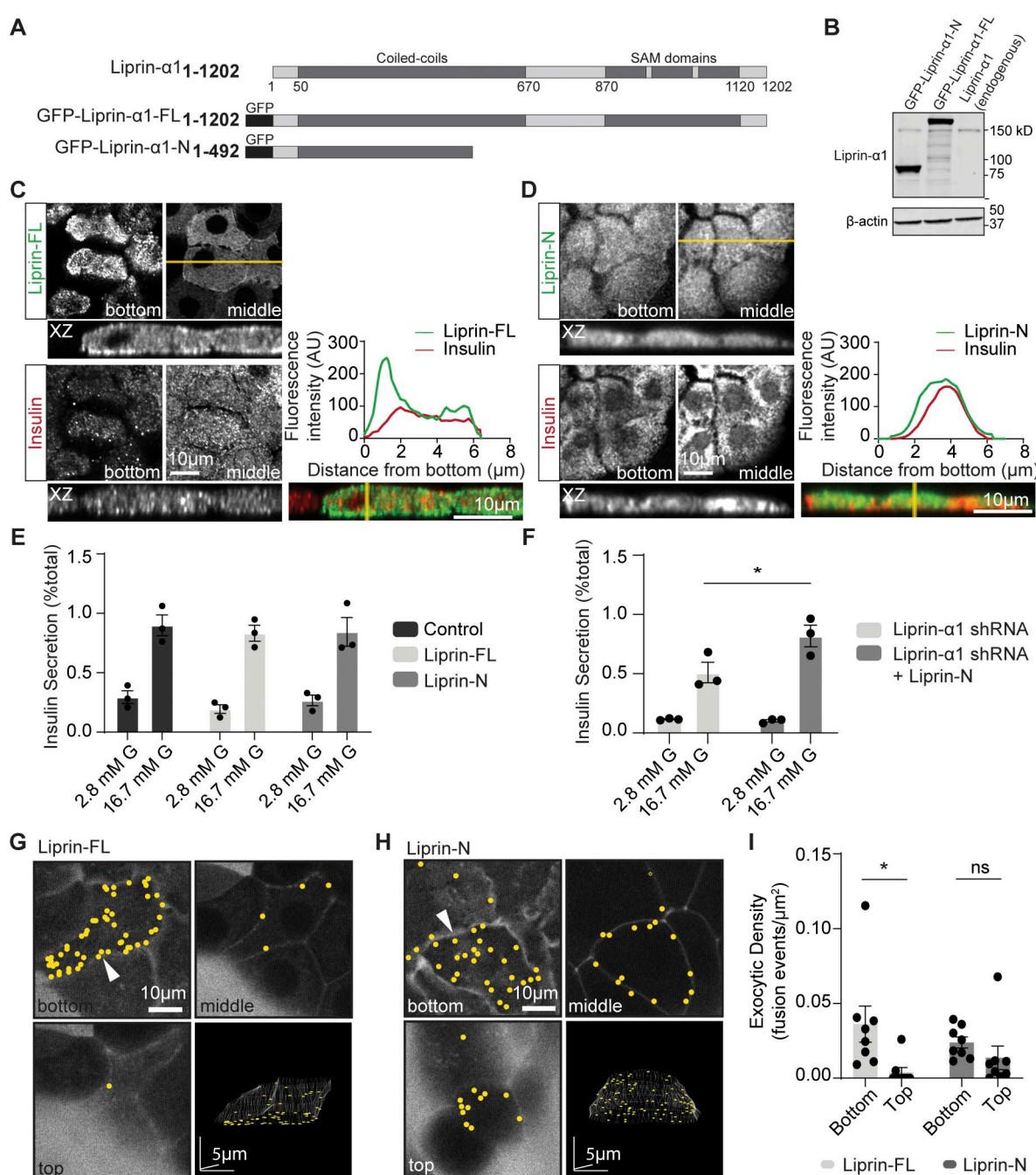

Figure 3. **The C terminus of liprin-α1 positions liprin-α1 and localizes insulin granule fusion to the ECM interface. (A)** Schematic of the domain organization of liprin-α1 and the residues encoding the GFP-tagged liprin-α1 constructs. Liprin-α1 consists of an N-terminal coiled-coil domain and a C-terminal region comprised of three sterile alpha motif (SAM) domains. **(B)** Mouse β cells were infected with adenovirus encoding each liprin-α1 construct. Western blot showing level of liprin-α1–FL and liprin-α1–N overexpression levels (~17X) compared with endogenous liprin-α1. **(C and D)** Immunofluorescence staining of liprin-α1 constructs (green) in isolated mouse β cells (insulin; blue) grown on coverslips coated with laminin-511, at the laminin-cell interface (bottom) compared with the middle. A line scan plotting fluorescence intensity across an orthogonal section (XZ) shows local enrichment of liprin-FL, but not liprin-N, at the laminin-cell interface. **(E and F)** Glucose-stimulated insulin secretion, normalized to total cellular insulin content, was comparable in β cells overexpressing GFP (control), liprin-FL, and liprin-N. **(F)** Expression of the N-terminal construct after liprin-α1 knockdown rescued secretion (for each condition, $n$ = 3 animals; two-way ANOVA followed by Tukey's multiple comparison, *: $P < 0.05$). **(G–I)** Isolated β cells overexpressing liprin-FL and liprin-N, cultured on laminin-511, were imaged using live-cell two-photon microscopy. Granule fusion was biased toward the coverslip in β cells expressing liprin-FL but not liprin-N. All scale bars: 10 µm, unless specified. All data are shown as mean ± SEM. Source data are available for this figure: SourceData F3.

consistent with the distribution of native liprin-α1 (see Fig. 1 A). In contrast, GFP-liprin-N was not enriched at the β cell–ECM interface and instead was evenly distributed throughout the cell (Fig. 3 D), supporting the idea that the C terminus is essential for locating liprin-α1 to the sites of focal adhesions that form at the β cell–ECM interface.

Overexpression of GFP-liprin-FL and GFP-liprin-N did not alter glucose-induced insulin secretion (Fig. 3 E). Moreover, liprin-N completely rescued glucose-induced secretion following liprin-α1 knockdown (Fig. 3 F), indicating it is necessary and sufficient for secretion in β cells, consistent with similar work in neurons (Chia et al., 2013; Taru and Jin, 2011). The loss of localization of GFP-liprin-N to the β cell–ECM interface (Fig. 3 D) suggested this might impact the localization of insulin granule fusion, which we tested using the live-cell 3D assay for granule fusion. In β cells cultured on laminin-511–coated coverslips and stimulated with high glucose (16.7 mM) for 15 min, we observed a significant bias in the number of granule fusion events at the β cell–ECM interface in cells overexpressing GFP-liprin-FL but not in cells expressing GFP-liprin-N (Fig. 3, G–I).

We conclude liprin-α1 has an essential role in regulating in glucose-induced insulin secretion. It appears that the N terminus alone is sufficient for this role but requires C-terminal interactions to localize to the ECM interface. Our data with high K+ stimulation demonstrate that liprin-α1 acts upstream of granule docking —granules already docked can fuse. Liprin-α1 knockdown disrupts the localization of granule fusion, suggesting liprin is part of a mechanism that locally positioning granules prior to docking and that this process(es) is under glucose control.

### Liprin-α1 assembles in clusters at the β cell–ECM interface

To further investigate the action of liprin-α1, we looked more closely at the subcellular organization of liprin-α1 at the β cell–ECM interface. In isolated mouse β cells cultured on laminin-511–coated coverslips, we used fixed cell immunofluorescence staining of liprin-α1. In low glucose (2.8 mM), we observed a punctate distribution of liprin-α1 across the ECM interface (Fig. 4 A). In cells fixed after 20 min of high glucose (16.7 mM) stimulation, the puncta increased in intensity (Fig. 4, B and D) but not after high K+ stimulation (Fig. 4, C and D). The results show that liprin-α1 is present as puncta at the β cell–ECM interface and that glucose stimulation specifically enriches liprin-α1 in these puncta.

### Live-cell super-resolution microscopy reveals dynamic liprin-α1 clusters at the β cell–ECM interface

To observe the dynamics of liprin-α1 clusters in more detail, we employed live-cell super-resolution spatial array confocal microscopy with β cells expressing GFP–liprin-α1. Characterization of GFP–liprin-α1 cluster size and density are not significantly different from native liprin-α1 and are provided in Fig. S2 (average size 0.11 μm²). Isolated β cells, cultured on laminin-511–coated coverslips, were imaged at the β cell–ECM interface and stimulated with high glucose (16.7 mM). Continuous recording of over 50 min showed the dynamic nature of these liprin-α1 clusters at the ECM interface, with clusters spontaneously

appearing and disappearing (Fig. 4, E–H). Furthermore, in response to glucose stimulation, we saw an increase in both the number of liprin-α1 clusters as well as their brightness (Fig. 4, F, G, and I–L), providing strong evidence of a glucose-dependent liprin-α1 translocation to the ECM interface.

### Liprin-α1 clusters are closely associated with sites of insulin granule fusion

In neurones, liprin-α1 clustering is essential for presynaptic AZ formation and enhancement of synaptic vesicle release (Emperador-Melero et al., 2021). We hypothesized that β cell liprin-α1 clusters might similarly be implicated in the localization of insulin granule fusion. We turned to our live-cell two-photon granule fusion assay to investigate the relationship between liprin-α1 and sites of insulin exocytosis. β cells expressing GFP–liprin-α1 were stimulated with 16.7 mM glucose, and granule fusion events were identified in time and space in relation to liprin-α1 (Fig. 5, A and B). Analysis of the distance of each granule fusion event to its nearest liprin-α1 cluster showed the preferential fusion of granules colocalized or immediately near these liprin-α1 clusters (Fig. 5, C and D), suggesting a role for liprin-α1 in positioning insulin granules to specific membrane sites at the ECM interface.

Even though our evidence indicates that liprin-α1 acts on granules prior to docking, we might expect to see an effect on granule positioning in the submembrane domain at the β cell–ECM interface. To study if this was the case, we used STED microscopy to resolve and count individual insulin granules at the ECM interface in isolated β cells cultured on laminin-coated coverslips for control and liprin-α1 knockdown. We observed no significant effect of liprin-α1 knockdown on the number of granules in this region (Fig. S3). Indeed, even in control cells stimulated with 16.7 mM glucose (Fig. S3), we saw no effect on the number of granules in the submembrane region. These findings are consistent with the observed very low number of fusing granules (typically <10 every minute) compared with the very high number of granules we see in this region (>1,000).

We conclude that liprin-α1 is acting locally to tether granules prior to docking. We next turned to co-IP and in vitro–binding assays to identify liprin-α1–binding partners and investigate how liprin-α1 might be associating with insulin granules.

### Liprin-α1 assembles in a presynaptic-like complex in MIN6 beta cells that interacts with insulin granules via β2-syntrophin

MIN6 cells were infected with adenovirus encoding GFP–liprin-α1 or GFP as a control. The GFP-tagged fusion proteins were co-immunoprecipitated with anti-GFP nanobody-conjugated beads. The immunoprecipitated samples were then subjected to bottom-up proteomics with data-independent acquisition (DIA) and protein identification and quantification using DIA-NN (Demichev et al., 2019). IPs were performed from both basal (2.8 mM glucose) and stimulated (16.7 mM glucose) conditions; however, no significant differences were observed between these two conditions (Data S1). Thus, the results from both conditions were pooled. Using SAINTexpress interactome analysis (Choi et al., 2011), we identified 49 proteins significantly (using the Bayesian false discovery rate) interacting with liprin-α1 (Fig. 6 A,

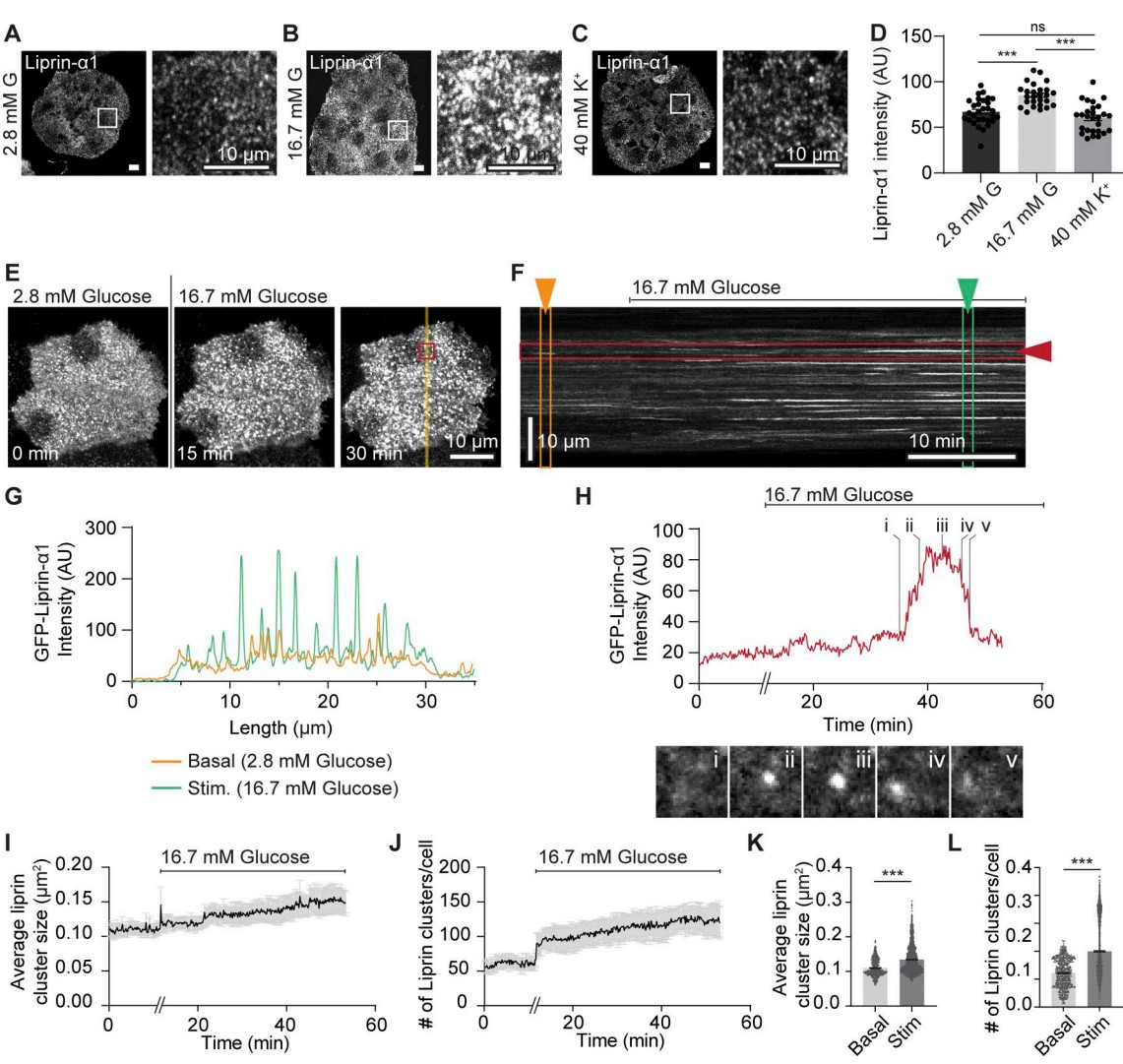

Figure 4. **Liprin-α1 assembles in glucose-dependent clusters at the β cell–ECM interface. (A–D)** Isolated β cells cultured on laminin-511 were stimulated with 16.7 mM glucose or 40 mM K⁺ and fixed and immunostained for endogenous liprin-α1. In 2.8 mM glucose conditions, liprin-α1 showed punctate distribution at the β cell–laminin interface. These liprin-α1 puncta showed increased fluorescence intensity after 16.7 mM glucose stimulation but not 40 mM K⁺ stimulation ($n = 3$ animals, one-way ANOVA followed by Tukey's multiple comparisons test). **(D–G)** Live-cell super-resolution spatial array confocal microscopy in β cells expressing GFP–liprin-α1, imaged at the β cell–laminin interface. **(D)** Snapshot of cells at 0 min (2.8 mM glucose), 15 min (16.7 mM glucose), and 30 min (16.7 mM glucose). **(B)** A kymograph showing liprin fluorescence over time along a line (indicated in yellow in Fig. 4 D) shows dynamic changes in liprin clusters, appearing and disappearing from the β cell–laminin interface over time. Clusters appear brighter and more abundant after glucose stimulation compared with before stimulation, (F) also apparent in line scans taken before (orange) and after (green) glucose stimulation. **(G)** Fluorescence changes over time of a region of interest (indicated in red in Fig. 4, D and E) shows the transient appearance of a single GFP–liprin-α1 cluster. **(H–L)** GFP–liprin-α1 cluster size and abundance (number of clusters per cell) tracked over time. Both size of clusters and number of clusters per cell increased after glucose stimulation ($n = 3$ animals, Student's *t* test, unpaired, equal variance).

Fig. S4, and Data S1). Analysis of these proteins revealed several predicted liprin-α1–binding partners, including PTPRF (LAR) (Serra-Pagès et al., 1995) (Fig. 6, A and B) and the synaptic signalling and scaffolding protein GIT1 (Ko et al., 2003a) (Fig. 6 A, Fig.S4, and Data S1). Furthermore, liprin family proteins liprin-α2 and liprin-β1, which have not been previously identified in β cells, as well as liprin-α1 itself, were all significantly immunoprecipitated (Fig. 6, A and B), likely reflecting liprin dimerization and oligomerization (Astro et al., 2016; Liang et al., 2021; Taru and Jin, 2011).

We predicted that liprin-α1 might interact with insulin granules *via* RIMs, the granule-associated Rab3-interacting

molecules, which are known to mediate vesicle docking and priming in both neurones (Han et al., 2011) and β cells (Iezzi et al., 2000; Yasuda et al., 2010). Furthermore, it is well established that RIMs interact with liprin-α3, the predominant liprin isoform in neurones, to form presynaptic scaffold complexes at the AZ (Schoch et al., 2002). Surprisingly, however, we did not identify RIMs as significant liprin-α1 interactors (Fig. 6 B), perhaps due to differences in liprin-α isoforms or cell types. Notably, however, we identified the insulin granule–associated protein SNTB2 (β2-syntrophin) (Schubert et al., 2010) as a liprin-α1–binding partner (Fig. 6 B), known to regulate secretory granule mobility by linking

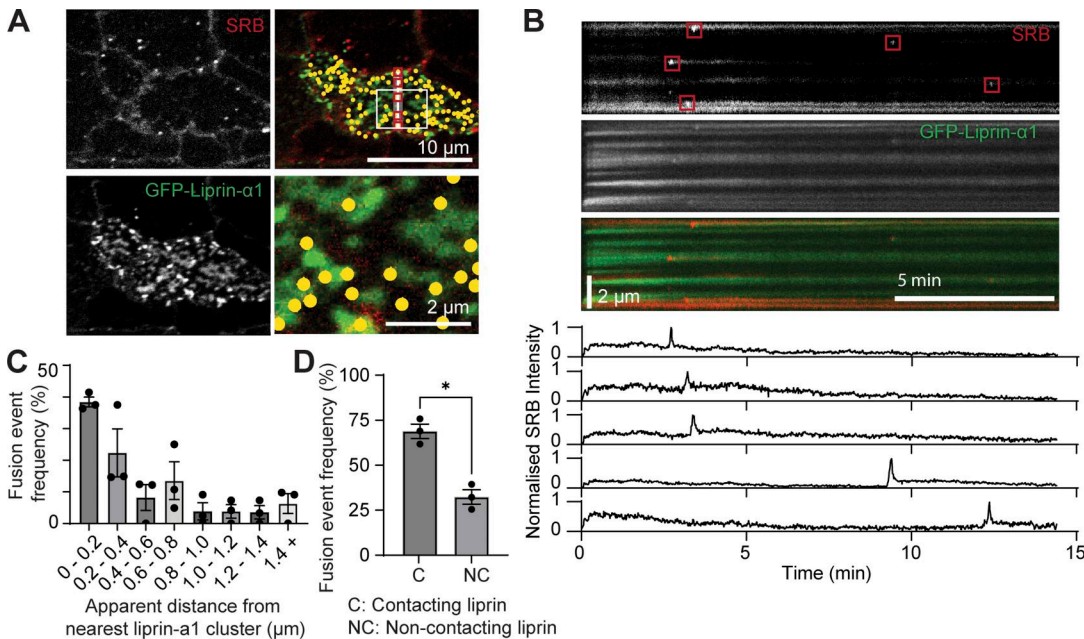

Figure 5. **Live-cell microscopy demonstrates preferential granule fusion near liprin-α1 clusters at the β cell–ECM interface. (A)** Live-cell two-photon imaging of isolated β cells expressing GFP–liprin-α1 (green) bathed in SRB (red). Cells were stimulated with 16.7 mM glucose for 15 min to induce granule fusion, identified in time and space, and marked with yellow dots. **(B)** A kymograph showing fluorescence over time along a line (indicated in white in Fig. 5 A) shows five example granule fusion events as small sudden bright flashes of SRB, also shown as sudden peaks in SRB intensity within a region of interest drawn over the fusing granules. Granule fusion sites overlap with regions of cell occupied by liprin-α1 clusters. **(C)** Frequency of granule fusion events in relation to distance from their nearest liprin-α1 neighbor. Measurements represent center-to-edge distances (center of a granule fusion event to edge of nearest GFP–liprin-α1 cluster) (three separate β cell clusters imaged, from three animals). **(D)** Percentage of granule fusion events contacting GFP–liprin-α1 plotted as a histogram, where contact is defined as a granule fusion event with any degree of colocalization with GFP–liprin-α1 (n = 3 animals, paired two-tailed Student's t test). All data are shown as mean ± SEM.

granules to the actin cytoskeleton (Ort et al., 2001; Schubert et al., 2010).

To validate these interactions, we performed co-IP using MIN6 cells that overexpressed GFP–liprin-α1 (or GFP control) followed by immunoblotting, confirming the binding of GFP–liprin-α1 with endogenous liprin-α1, liprin-β1, and β2-syntrophin (Fig. 6 C). Furthermore, to validate these interactions in the native system, we performed co-IP with natively expressed liprin-α1 and again showed binding to liprin-β1 and β2-syntrophin (Fig. 6 D). As further validation, we performed immunofluorescence staining for β2-syntrophin in mouse β cells within a pancreatic slice. While β2-syntrophin showed diffuse distribution across the cell cytosol reminiscent of insulin granule distribution, we also observed significant enrichment at the β cell–capillary interface (labelled with laminin) like liprin-α1, apparent when comparing the relative fluorescence intensities in regions of interest places at the basal (vascular), lateral (between the cells), and apical (abvascular) regions of the cells (see [Gan et al., 2017] for details of β cell polarity) and with a Pearson correlation analysis of liprin-α1 with β2-syntrophin distribution (Student's t test P < 0.001 comparing overlay with a 90° rotation of one of the images, n = 8 islets from 3 animals) (Fig. 6 E). Using isolated β cells cultured on laminin-511–coated coverslips, we also observed colocalization of β2-syntrophin with liprin-α1 clusters across the ECM interface (Fig. 6 F). Taken together, these data support a model where liprin-α1 assembles in a β cell

presynaptic-like complex that interacts with insulin granules via β2-syntrophin to localize granules at the ECM interface.

## Discussion

The localization of insulin granule fusion to the ECM interface (Bonner-Weir, 1988; Jevon et al., 2022; Low et al., 2014) is likely to involve multiple mechanisms regulating both signalling pathways and granule behavior. Here, we provide evidence that granule positioning, prior to docking, is controlled by liprin-α1. We show that three distinct forms of stimulation all lead to exocytosis localized to the ECM interface. The stimuli include global uncaging of Ca²⁺, which indicates that localized granule fusion is not dependent on local Ca²⁺ entry. Knockdown of liprin-α1 impairs localized granule fusion and reduces both phases of glucose-induced insulin secretion. Moreover, liprin-α1 assembles in clusters across the ECM interface. The size and number of clusters are dynamically regulated by glucose stimulation, and the sites of insulin granule fusion are closely linked to the liprin clusters. Analysis of the liprin-α1 interactome revealed protein–protein interactions reminiscent of the neuronal presynaptic scaffold complex and specifically identified β2-syntrophin as a potential link with insulin granules. Taken together, we propose a model where liprin-α1 localizes granules prior to docking, and this process is an additional step in the stimulus–secretion pathway of glucose-induced secretion.

Figure 6.   **Liprin-α1 assembles in a presynaptic-like complex in MIN6 β cells and interacts with insulin granules via β2-syntrophin. (A)** MIN6 cells expressing GFP (control) or GFP–liprin-α1 were incubated in basal (2.8 mM glucose) and stimulated (16.7 mM glucose) conditions. Cells were lysed for IP with anti-GFP nanobody-conjugated beads. Immunoprecipitates were subjected to bottom-up proteomics with DIA and protein identification and quantification using DIA-NN. Immunoprecipitates from basal and stimulated groups were pooled, and significant interactors were identified with SAINTexpress ($n$ = 6 GFP control, $n$ = 6 GFP–liprin-α1) and plotted with log10 average intensity on the x axis and log$_2$ fold change of GFP–liprin-α1 over GFP control on the y axis. Each point represents an individual protein; red points represent statistical significance Bayesian false discovery rate (BFDR). **(B)** Box-and-whisker plots showing median LFQ intensities and 1.5 times the interquartile range for specific proteins of interest. B, basal; S, stimulated. **(C)** Anti-GFP immunoprecipitates were also analyzed by immunoblotting, showing pull-down of liprin-β1 and β2-syntrophin with GFP–liprin-α1. Band intensities are plotted as a bar graph, normalized to GFP control. **(D)** Co-IP of native liprin-α1 with liprin-β1 and β2-syntrophin in MIN6 β cells. Cells were lysed for IP with protein A/G magnetic beads, and the immunoprecipitates were analyzed by immunoblotting using anti-liprin-α1, anti–liprin-β1, and anti–β2-syntrophin antibodies. Band intensities are plotted as a bar graph, normalized to IgG. **(E)** Representative immunofluorescence of an islet within a pancreatic slice. Liprin-α1 (red) and β2-syntrophin (green) are both enriched at the β cell–ECM interface (laminin; blue), also seen in a histogram showing relative fluorescence intensity at the β cell basal (capillary), apical, and lateral regions (49 cells, 9 islets across 3 animals), and a line scan across a region of interest. Quantification of colocalization between liprin-α1 and β2-

syntrophin using Pearson's correlation coefficient between the two channels. 90° indicates a 90° rotation of one of the two analyzed channels before analysis. **(F)** Immunofluorescence staining of liprin-α1 and β2-syntrophin in isolated dispersed β cells. β2-syntrophin is present and colocalized in liprin-α1 clusters across the β cell/laminin interface. Source data are available for this figure: SourceData F6.

## Presynaptic-like mechanisms and the control of insulin secretion in the β cell

Previous work has identified aspects of the control of insulin secretion that resemble presynaptic mechanisms. For example, the β cell secretory domain is characterized by an enrichment of presynaptic scaffold proteins (Low et al., 2014) and the local activation of Ca$^{2+}$ channels (Jevon et al., 2022). Furthermore, ELKS is also known to interact with Ca$^{2+}$ channels to facilitate polarized Ca$^{2+}$ influx (Ohara-Imaizumi et al., 2019), and in MIN6 cells, ELKS clusters were spatially linked to granule fusion (Ohara-Imaizumi et al., 2005). However, whether presynaptic scaffold proteins exist as a complex that recruits, positions, and locally regulates components of the β cell secretory machinery is not clear. Here, we provide the first evidence demonstrating a role for liprin-α1 in positioning granules to the ECM interface, consistent with one of the key functions of the neuronal presynaptic complex. Moreover, we perform the first comprehensive interactome analysis of liprin-α1 in the β cell to directly demonstrate the existence of a presynaptic-like complex in the β cell.

Our data show that liprin-α1 knockdown does not affect high K$^+$-induced insulin secretion but does disrupt the localization of granule fusion. Since high K$^+$ stimulation leads a short-lasting response and the fusion of docked granules (Shibasaki et al., 2007), our data suggest that liprin-α1 plays a role upstream of docking. In control experiments with high K$^+$ stimulation, granule fusion is localized to the ECM interface; this is disrupted with liprin-α1 knockdown, showing that granules can still dock and fuse, but the spatial constraint imposed by liprin-α1 is lost.

While we do not understand the mechanistic detail, this work suggests that liprin-α1 either directly interacts with granules prior to docking or indirectly regulates other processes to spatially control the translocation of granules to the cell membrane. The former process is supported by our finding of β2-syntrophin, a granule surface protein, as a binding partner. For the latter, we know that liprin can form complexes with focal adhesions (Bouchet et al., 2016), directly supported by our finding of liprin β1 as a binding partner, and in turn, focal adhesions are hubs for microfilaments and microtubules, both of which are required for granule positioning (Bracey et al., 2020).

In contrast to high K$^+$, glucose-stimulated insulin secretion is affected by liprin-α1 knockdown. We show that liprin-α1 forms clusters at the ECM interface. The number and size of the clusters are under glucose control, and the clusters are a focus for sites of granule fusion. The observation that liprin-α1 clusters do not change with high K$^+$ stimulation suggests that a rise in cytosolic Ca$^{2+}$ (which occurs for both high K$^+$ and glucose stimulation) is not a sufficient trigger for translocation. Distinct pathways regulating glucose-dependent secretion are explicit in the trigger and amplification steps in the model for secretory control of Henquin (Henquin, 2000). Our data suggest that liprin-α1 is part of the amplification pathway. The mechanism is

unknown, but cytoskeletal changes might be involved in the formation of liprin-α1 clusters, and there is abundant evidence for glucose-dependent control of microtubule (Trogden et al., 2019) and microfilament (Wang and Thurmond, 2009) structures.

## How might liprin-α1 be located to the ECM interface?

In neurones, the presynaptic complex is positioned using transsynaptic cues like neurexins (Südhof, 2008) and LAR (Emperador-Melero and Kaeser, 2020). In the absence of a postsynaptic domain, alternate environmental cues and anchoring mechanisms must be present in β cells. Our previous work shows that liprin-α1 positioning is linked to local activation of the integrin/focal adhesion kinase pathway (Jevon et al., 2022). Inhibiting integrin activation disrupts the localization of liprin-α1 and ELKS (Jevon et al., 2022) and also leads to the mistargeting of insulin granule fusion (Gan et al., 2018). The mechanism linking liprin-α1 to activated integrins remains unknown. However, studies in other cell types, including fibroblasts, suggest that liprin and ELKS interact with focal adhesion–associated proteins KANK1 and Ll5β (Bouchet et al., 2016) to form cortical microtubule stabilization complexes (CMSCs) near focal adhesions (Fourriere et al., 2019; Grigoriev et al., 2007; Lansbergen et al., 2006; Stehbens et al., 2014). Recently, both KANK1 and Ll5β have been identified in mouse and human islets (Noordstra et al., 2022), supporting the idea that CMSCs might also be present in β cells. Notably, liprin-β1, which we now show interacts with liprin-α1, is an essential component of CMSCs that serves as a physical linkage between CMSCs and focal adhesions (Bouchet et al., 2016; van der Vaart et al., 2013) and provides further support for a mechanism that holds liprin-α1 close to focal adhesions *via* CMSCs.

We also identify LAR, which has not previously been shown in β cells, as a liprin-α1–binding partner. LAR is a transmembrane protein that interacts with ECM proteins laminin and nidogen (O'Grady et al., 1998), both components of the islet vascular basement membrane (Virtanen et al., 2008), and could therefore also serve as a link between liprin-α1 and the ECM interface. Further work is required to understand how the liprin complex might be positioned to the ECM interface to regulate targeted secretion.

## The β cell presynaptic-like complex

Liprin-α1 has a broad tissue distribution (Serra-Pagès et al., 1998) and shares a common structure with other liprin isoforms (Serra-Pagès et al., 1998) as well as common protein interactors (Ko et al., 2003a; Ko et al., 2003b; Schoch et al., 2002; Serra-Pagès et al., 1998; Wei et al., 2011). There are, however, known distinctions among the isoforms, for example, liprin-α2 interacts with CASK but liprin-α1 does not (Wei et al., 2011).

Here, we show that liprin-α1 binds to itself, liprin-β1, and liprin-α2, indicating that oligomerization is important.

However, unlike past work, we fail to observe ELKS (Ko et al., 2003b) or RIM (Schoch et al., 2002) binding to liprin-α1. This likely reflects tissue, species, and/or methodological differences, as these previous studies utilized yeast two-hybrid screens and GST pull-downs in rat brain tissue, HEK293T, and COS cells. Importantly, we identify numerous AZ-associated proteins interacting with liprin-α1, including GIT1 (Ko et al., 2003a) and LAR (Serra-Pagès et al., 1995), supporting the idea of presynaptic-like regulation (Deng and Thorn, 2022). Consistent with previous reports, we also identify five members of the protein phosphatase 2A complex (Ripamonti et al., 2022; Xie et al., 2021) and 7 paralogs of 14-3-3 proteins, which are known to interact with liprin-β1 (Segal et al., 2023) and other presynaptic proteins (Schröder et al., 2013). Together, these data suggest that liprin-α1 is central to a wider complex that is the focus for phosphorylation.

We speculate that this presynaptic complex interacts with components of the distal stages of granule docking, priming, and fusion. Our data show that liprin-α1 clusters are intimately linked with sites of granule fusion, and similar data in MIN6 cells show fusion occurs close to ELKS clusters (Ohara-Imaizumi et al., 2005). Since granule fusion is correlated with dynamic recruitment of SNAREs (Gandasi and Barg, 2014), we expect that the presynaptic complex plays a role in this recruitment. However, specifically for liprin-α1, our data strongly point to a role prior to docking and priming. Our identification of liprin-α1 binding to β2-syntrophin now provides a potential direct link to granule positioning. β2-syntrophin binds the insulin granule protein ICA512 and interacts with F-actin to control granule movement in the β cell cortex (Ort et al., 2001; Schubert et al., 2010). We also show that β2-syntrophin is enriched at the ECM interface and colocalized within liprin-α1 clusters. Clearly more work is required to elucidate whether the liprin-α1/β2-syntrophin interaction impacts on granule and F-actin binding to regulate granule targeting.

Previous work has identified liprin-α1 in human β cells (Cottle et al., 2021), where it is also enriched at the capillary interface, suggesting a similar function to mouse β cells. Species differences might give rise to different isoforms and additional mechanisms of control, but the broad principles of presynaptic-like control of insulin secretion are likely to be preserved in human β cells.

### Broader relevance of our findings

Subcortical granule positioning is essential in any secretory cell and is a required step prior to granule docking, priming, and exocytosis (Deng and Thorn, 2022). This is particularly important for cells where granule fusion is regionally confined, with the best example being the presynaptic domain, where scaffold proteins position neurotransmitter vesicles prior to docking (Südhof, 2012). Our findings demonstrate that liprin-α1 serves this function in pancreatic β cells, adding a significant new level of complexity to the control of insulin secretion. Importantly, we provide evidence that this mechanism is specifically controlled by glucose. It is well-known that glucose directly triggers insulin secretion through membrane depolarization, but glucose also acts through unknown mechanisms to increase the secretory

capacity of β cells (Henquin, 2009). We demonstrate that liprin-α1 is a key component in a glucose-dependent mechanism of granule positioning and, together with the broader presynaptic-like protein complex, highlights a new and critical step in β cell stimulus–secretion coupling and could have important implications in refining cell-based therapies for type 1 diabetes.

In conclusion, our work demonstrates a novel presynaptic model for the control of glucose-dependent insulin secretion where liprin-α1 assembles in a presynaptic-like complex to control the localization of insulin granule fusion.

## Materials and methods

### Antibodies and reagents

The following antibodies were used: anti-beta1 laminin (Cat #MA5-14657; Thermo Fisher Scientific; RRID: AB_10981503), anti-insulin (Cat #A0564; Dako; RRID: AB_726362), Anti-liprin alpha1 (Cat #14175-1-AP; Proteintech), anti-SNTB2 (Cat #MA1-745; Thermo Fisher Scientific; RRID: AB_2191939), anti-Ppfibp1 (Cat #PA5-51663; Thermo Fisher Scientific; RRID: AB_2645815), Alexa Fluor 488 goat anti-guinea pig (Cat #A11073; Thermo Fisher Scientific; RRID: AB_2534117), Alexa Fluor 546 Donkey anti-mouse (Cat #A10036; Thermo Fisher Scientific; RRID: 2534012), Alexa Fluor 594 Donkey anti-rabbit (Cat #R37119; Thermo Fisher Scientific; RRID: AB_2556547), and Alexa Fluor 633 Got anti-rat (Cat #A21094; Thermo Fisher Scientific; RRID: 2535749). The following reagents were used: Liberase TL Research Grade (SKU 5401020001; Sigma-Aldrich), Histopaque-1119 (SKU 11191; Sigma-Aldrich), TrypLE express enzyme (SKU 10771; Sigma-Aldrich), RPMI 1640 Medium (Cat #11875085; Thermo Fisher Scientific), DMEM (Cat #11995073; Thermo Fisher Scientific), Fetal Bovine Serum (USDA APPD Origin, Cat #10437028; Thermo Fisher Scientific), Penicillin-Streptomycin (Cat #15140122; Thermo Fisher Scientific), 2-Mercaptoethanol (Cat #21985023; Thermo Fisher Scientific), Human recombinant laminin 511 (LN511; BioLamina), Fura-2 AM cell permeant (Cat #F1221; Thermo Fisher Scientific), sulforhodamine B (Cat #S1307; Thermo Fisher Scientific), NP-EGTA AM (o-Nitrophenyl EGTA AM, cell permeant, Cat #11529156; Thermo Fisher Scientific), Soybean Trypsin Inhibitor (Cat #17075029; Thermo Fisher Scientific), Water (Cat #FSBW6-4; Thermo Fisher Scientific), Acetonitrile (Cat #FSBA955-4; Thermo Fisher Scientific), Ethyl acetate (Cat #109623; Merck Millipore), Triscarboxyethylphosphine (TCEP, Neutral pH solution, Cat #77720; Thermo Fisher Scientific), Chloracetamide (SKU C0267; Sigma-Aldrich), Trypsin (Cat #90059; Thermo Fisher Scientific), Sodium deoxycholate (SKU D6750; Sigma-Aldrich), and Styrenedivinylbenzene reversed-phase sulfonate (Cat #66886-U; Merck Millipore).

### Animal husbandry

Male C57BL/6 was housed at the Charles Perkins Centre Laboratory Animal Services facility in a specific pathogen-free environment, at 22°C with 12-h light cycles. All mice were fed a standard chow diet (7% simple sugars, 3% fat, 50% polysaccharide, 15% protein [wt/wt], and energy 3.5 kcal/g). Mice (8–12 wk old) were humanely killed according to local animal ethics

procedures (approved by the University of Sydney Research Integrity and Ethics Administration Committee, project #2023/2300).

### Islet isolation and dispersion
Isolated mouse islets were prepared according to a standard method that utilizes collagenase enzymes for digestion and separation from exocrine pancreatic tissue (Hoppa et al., 2009). In brief, a Liberase solution was prepared in unsupplemented RPMI-1640 (0.5 U/ml). Pancreases were distended by injection of 2 ml of ice-cold Liberase solution via the pancreatic duct, dissected, and placed into sterile tubes in a 37°C water bath for 14 min. Isolated islets were separated from the cell debris using a Histopaque density gradient. Isolated islets were cultured for 16 h (37°C, 95/5% air/$CO_2$) in RPMI-1640 culture medium supplemented with 10% fetal bovine serum and 1% penicillin-streptomycin before islet dispersion. Islets were incubated with TrypLE in a 37°C water bath for 4.5 min. Islet cells were resuspended in islet media and dispersed onto laminin-511–coated coverslips.

### Islet slices
Sectioning of unfixed pancreatic tissue was performed as described by Gan et al. (2018), Huang et al. (2011). Pancreatic sections (200-µm thick) were sliced using a vibratome and cultured for 16 h (37°C, 95/5% air/$CO_2$) in RPMI-1640 culture medium supplemented with 10% fetal bovine serum, 1% penicillin-streptomycin, and 100 µg/ml soybean trypsin inhibitor.

### Cell lines
Mouse insulinoma (MIN6) cells were cultured in DMEM supplemented with 15% FBS, 1% penicillin-streptomycin, and 0.05 mM 2-mercaptoethanol under standard culture conditions (37°C, 95/5% air/$CO_2$). Culture media was changed every second day, and the cells were regularly passaged upon reaching 80 % confluency using TrypLE express enzyme. All MIN6 cells used in this study were at passages <30 to maintain a normal glucose-stimulated insulin secretion phenotype (Cheng et al., 2012).

### Adenoviruses
Where indicated, β cells or MIN6 cells were infected with adenovirus containing GFP-scrambled shRNA, GFP-(mouse) liprin-α1 shRNA, GFP, GFP-(mouse) liprin-α1 aa 1–492, GFP-(mouse) liprin-α1 aa 493–1202, or GFP-(human) liprin-α1 and incubated for 72 h before experiments (Vector Biolabs).

### Protein coating
Coverslips were coated with laminin-511 (5 µg/ml) overnight at 4°C. Coated coverslips were briefly rinsed in sterile PBS before islet dispersion.

### Immunofluorescence
Samples were fixed with 4% paraformaldehyde in PBS for 15 min at room temperature. Immunofluorescence was performed as described by Meneghel-Rozzo et al. (2004). Tissues were incubated in blocking buffer (3% BSA, 3% donkey serum, and 0.3%

Triton X-100) for a minimum of 1 h at room temperature, followed by primary antibody incubation at 4°C overnight and secondary antibody incubation for 4 h (slices) or 1 h (dispersed cells) at room temperature. All primary and secondary antibodies were used at a 1:200 dilution. DAPI (100 ng/ml, Cat #D3571; Thermo Fisher Scientific) was with the secondary antibodies. Samples were mounted using ProLong Diamond Antifade Mountant and imaged on a Leica SP8 confocal microscopy with a 63× oil immersion objective. Images were analyzed using FIJI ImageJ.

### Live-cell imaging
3D live-cell multiphoton imaging was performed on a custom-made Olympus two-photon microscope. For granule fusion assays, cells were incubated in 2.8 mM glucose Krebs–Ringer bicarbonate HEPES buffer (KRBH, 120 mM NaCl, 4.56 mM KCl, 1.2 mM $KH_2PO_4$, 1.2 mM $MgSO_4$, 15 mM $NaHCO_3$, 10 mM HEPES, 2.5 mM $CaCl_2$, and 0.2% BSA, pH 7.4) containing extracellular dye (SRB, 8 mM).

We used a custom-made two-photon microscope with a 60× oil immersion objective (NA 1.42, Olympus). Excitation was at 850 nm, and fluorescence emission was detected at 550–650 nm with a frame rate of 6 Hz. 3D images were collected at a frame rate of 6 Hz with z sections 2-µm apart. Images (resolution of 10 pixels/µm) were captured using ScanImage software (Pologruto et al., 2003) controlling custom hardware.

Cells were stimulated with 16.7 mM glucose KRBH or 40 mM $K^+$ KRBH containing 8 mM SRB, and exocytotic events were recorded as the entry of SRB into each fusing granule upon stimulation. Images were analyzed using FIJI ImageJ and Meta-Morph (Molecular Devices) software. 3D projections were created using IMOD (The Regents of the University of Colorado). Live-cell super-resolution imaging was performed using the Nikon AX/R with Spatial Array Confocal confocal-based super-resolution microscope. Time-lapse (6 frames/min) imaging was performed for 50 min using a high-speed resonant scanner in combination with the Nikon AX/R with Spatial Array Confocal detector. Cells were incubated in 2.8 mM glucose KRBH at 37°C and 5% $CO_2$, followed by stimulation with 16.7 mM glucose.

### Analysis of granule fusion events
Granule fusion events were measured from regions of interest (0.78 µm²) centered over individual granules. Traces were rejected if extensive movement was observed. For the 3D mapping of exocytosis, the first optical two-photon section focused at the cell-coverslip interface (within 1 µm of coverslip surface) was defined as the "bottom" plane. Subsequent optical sections (detecting exocytotic events >1 µm from the coverslip) were defined as "upper" planes. 2D granule fusion colocalization and proximity analyses (Fig. 5, C and D) were performed in ImageJ using the DiAna plugin (Gilles et al., 2017). GFP–liprin-α1 clusters were identified with the spot segmentation procedure using the following parameters: maxima detection: radius in xy axis 2, noise 5, threshold for maxima selection 100; parameters for Gaussian fit and threshold calculation: radius maximum 10, SD 5. After GFP–liprin-α1 segmentation, we performed center-edge distance analysis (distance from center of granule fusion event to edge of

nearest GFP–liprin-α1 cluster). Segmented GFP–liprin-α1 and circular ROIs (diameter 0.3 μm; the average size of an insulin granule) centered over the brightest SRB signal were used for object-based colocalization analyses using the DiAna plugin.

## Photoliberation of $Ca^{2+}$ from NP-EGTA
Dispersed β cells were loaded with 6 μM NP-EGTA, AM and 2 μM Fura-2, AM (for 1 h) in 2.8 mM glucose KRBH. An epifluorescence mercury light source provided high-intensity UV light to uncage $Ca^{2+}$ from NP-EGTA in a ~30-μm diameter field at the image plane. The duration of exposure to UV light was set to 100 ms by a computer-controlled shutter.

## Glucose-stimulated insulin secretion and homogeneous time resolved fluorescence insulin assay
Cells were incubated in 2.8 mM glucose KRBH for 1 h at 37°C and 5% $CO_2$ (pre-basal). Cells were washed and then incubated in 2.8 mM glucose KRBH again for 30 min (basal), collecting the supernatant. Cells were then stimulated with either 16.7 mM glucose KRBH or modified 2.8 mM glucose KRBH with reduced NaCl (100 mM) and high potassium (40 mM KCl), collecting the supernatant. All media and cells were kept at 37°C and 5% $CO_2$ for the duration of the assay. Cells were lysed at the end of the assay using ice-cold lysis buffer (1% NP-40, 300 mM NaCl, 50 mM Tris-HCl, pH 7.4, and protease inhibitor cocktail tablet) and sonicated. Supernatants and lysates were stored at –30°C prior to homogeneous time resolved fluorescence (HTRF) assay (HTRF Insulin Ultra-Sensitive Detection Kit, Cat #62IN2PEG; Revvity) performed according to manufacturer's instructions.

## Fura-2 $Ca^{2+}$ imaging
$Ca^{2+}$ imaging was performed using the Nikon Ti-E Spinning Disc Confocal microscope. β cells were incubated in 2 μM Fura-2, AM (for 1 h) in 2.8 mM glucose KRBH at 37°C and 5% $CO_2$, followed by 16.7 mM glucose KRBH for 15 min. $Ca^{2+}$ measurement and calibration were performed according to Grynkiewicz et al. (1985).

## IP
Mouse insulinoma (MIN6) cells were purchased from AddexBio (C0018008; AddexBio Technologies). MIN6 cells expressing GFP (control) or GFP–liprin-α1 were incubated in 2.8 mM glucose KRBH or 16.7 mM glucose KRBH for 2 h at 37°C and 5% $CO_2$. Cells were lysed in ice-cold lysis buffer (1% NP40, 10% glycerol, 137 mM NaCl, 25 mM Tris, 1 X cOmplete Protease Inhibitor Cocktail, and 1 × PhosStop phosphatase inhibitor, pH 7.4) and then sonicated over ice at 90% amplitude for 3:3-s pulses for 2 min on-time using a probe sonicator. Sonicated cell lysates were clarified by centrifugation at 18,000 × g for 10 min at 4°C, and the supernatant was collected. GFP-Trap agarose beads (Proteintech, RRID:AB_2631357) were washed twice with PBS. Clarified cell lysates were incubated with GFP-Trap beads (25 μl bead slurry/4 mg protein) at 4°C for 1 h with rotation. Beads were pelleted and washed twice with wash buffer (0.05% NP40, 10% glycerol, 137 mM NaCl, and 25 mM Tris, pH 7.4) and once with PBS. Bound proteins were eluted by boiling the beads in 4% SDC and 0.1 mM Tris-HCl, pH 8.0, for 5 min at 95°C with shaking.

For endogenous-binding studies, uninfected MIN6 cells were incubated in 16.7 mM glucose for 2 h at 37°C and 5% $CO_2$ KRBH. Cells were lysed in ice-cold lysis buffer (1% NP40, 10% glycerol, 137 mM NaCl, 25 mM Tris, 1 X cOmplete Protease Inhibitor Cocktail, and 1 × PhosStop phosphatase inhibitor, pH 7.4) and then sonicated over ice at 90% amplitude for 3:3-s pulses for 2 min on-time using a probe sonicator. Sonicated cell lysates were clarified by centrifugation at 18,000 × g for 10 min at 4°C, and the supernatant was collected. Clarified cell lysates (4 mg per IP) were incubated with 10 μg anti–liprin-α1, anti–β2-syntrophin, anti-mouse, or anti-rabbit IgG antibodies overnight at 4°C with rotation. 25 μl of Pierce Protein A/G Magnetic Beads (Cat #88802; Thermo Fisher Scientific) were added to each lysate mixture and incubated at 4°C for 1 h with rotation. Beads were pelleted and washed twice with wash buffer (0.05% NP40, 10% glycerol, 137 mM NaCl, and 25 mM Tris, pH 7.4) and once with PBS. Bound proteins were eluted by heating the beads in 2% SDS, 20 mM $NaPO_4$, 50 mM NaCl, and 10 mM TCEP for 10 min at 65°C with shaking.

## Immunoblotting
SDS-PAGE was performed on Novex Tris-Glycine Mini Protein Gels, 4–12%. After separation, proteins were transferred to Immobilon-FL PVDF Membranes. Membranes were blocked with SuperBlock Blocking Buffer for 10 min at room temperature, followed by an overnight incubation with rabbit anti–liprin-α1 (1:1,000), rabbit anti–liprin-β1 (1:1,000), or mouse anti-SNTB2 (1:1,000) at 4°C. After washing with TBST, membranes were incubated for 1 h at room temperature with anti-rabbit or anti-mouse IRDye secondary antibodies and visualized by a Licor Odyssey CLx system.

## Sample preparation and mass spectrometry–based interactome analysis
Trypsin digestion was performed as described previously (Harney et al., 2021). Briefly, IP samples were reduced with 10 mM TCEP and alkylated with 40 mM chloroacetamide at 95°C for 10 min. IP samples were then diluted to a final concentration of 1% SDC using water and digested with 400 ng MS-grade trypsin at 37°C for 16 h. The trypsin digest was stopped by adding an equal volume of 99% ethylacetate/1% TFA (49.5% ethyl acetate and 0.5% TFA final concentration, vol/vol). Sample cleanup using styrenedivinylbenzene reversed-phase sulfonate StageTips was performed as described previously (Harney et al., 2019). Dried peptides were reconstituted with 5% formic acid, sealed, and stored at 4°C until LC-MS/MS acquisition. Peptides were directly injected onto a 20-cm × 75-μm C18- (Dr. Maisch, Ammerbuch, Germany, 1.9 μm) fused silica analytical column with a 10-μm pulled tip, coupled online to a nanospray ESI source. Peptides were resolved using a NeoVanquish UHPLC (Thermo Fisher Scientific) over a gradient from 7% to 35% acetonitrile for 18 min with a flow rate of 300 nl min-1. Peptide ionization by electrospray occurred at 2.4 kV. An Exploris-480 mass spectrometer (Thermo Fisher Scientific) with HCD fragmentation was used for MS/MS acquisition. Spectra were obtained in a DIA using 15 variable isolation width DIA windows. Protein identification and quantification was performed using DIA-NN (Demichev et al., 2019). The DIA-NN output was

uploaded to the ProteomeXchange Consortium under the identifier PXD049219, username: reviewer_pxd049219@ebi.ac.uk, password: R3tuAIyu. For library generation, we use the combined UniProt mouse (Swiss-Prot and TrEMBL) databases that were downloaded on the 22nd of September 2023. Fully specific trypsin was set as the protease allowing for 1 missed cleavage and 1 variable modification. Protein N terminus acetylation and oxidation of methionine were set as variable modifications. Carbamidomethylation of cystine was set as a fixed modification. Remove likely interferences and match between runs were enabled. Neural network classifier was set to double-pass mode. Protein inference was based on genes. Quantification strategy was set to any LC (high accuracy). Cross-run normalization was set to RT dependent. Library profiling was set to smart profiling.

#### Quantification and statistical analysis

DIA-NN outputs for the interactome analysis were prepared for SAINTexpress (Choi et al., 2011) using R. GFP-expressing cells ($n = 6$) were used as negative controls. Prey proteins were considered significant if they passed the Bayesian FDR cutoff of <0.05.

#### Online supplemental material

The supplemental material in Figs. S1, S2, and S3 contain imaging data that support statements made in the main text. Fig. S4 extends the analysis of the mass spectrometry data with a STRING analysis. Data S1 (IP-MS SAINT).

#### Data availability

The data underlying Figs. 1, 2, 3, 4, and 5 are available in the published article and its online supplemental material. The data underlying IP and mass spectrometry analysis, shown in Fig. 6, are available, upon request, from the corresponding author.

## Acknowledgments

We acknowledge project funding obtained from the National Health and Medical Research Council (APP1128273, to Peter Thorn), Diabetes Australia (DART grant to Peter Thorn), and the Bioscientifica Trust (BT-000084, to Kylie Deng). Imaging was performed in the Centre for Microscopy and Microanalysis at the University of Sydney (ACMM, SMM). Open Access funding provided by University of Sydney.

Author contributions: Kylie Deng: conceptualization, data curation, formal analysis, investigation, methodology, visualization, and writing—original draft, review, and editing. Kitty Sun: investigation. Nicole Hallahan: investigation and supervision. Wan Jun Gan: methodology. Michelle Cielesh: investigation. Baharak Mahyad: data curation and formal analysis. Melkam A. Kebede: supervision and writing—review and editing. Mark Larance: data curation, formal analysis, investigation, methodology, resources, supervision, and writing—review and editing. Peter Thorn: conceptualization, data curation, formal analysis, funding acquisition, investigation, methodology, project administration, resources, supervision, validation, visualization, and writing—original draft, review, and editing.

Disclosures: The authors declare no competing interests exist.

Submitted: 1 November 2024

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

# Supplemental material

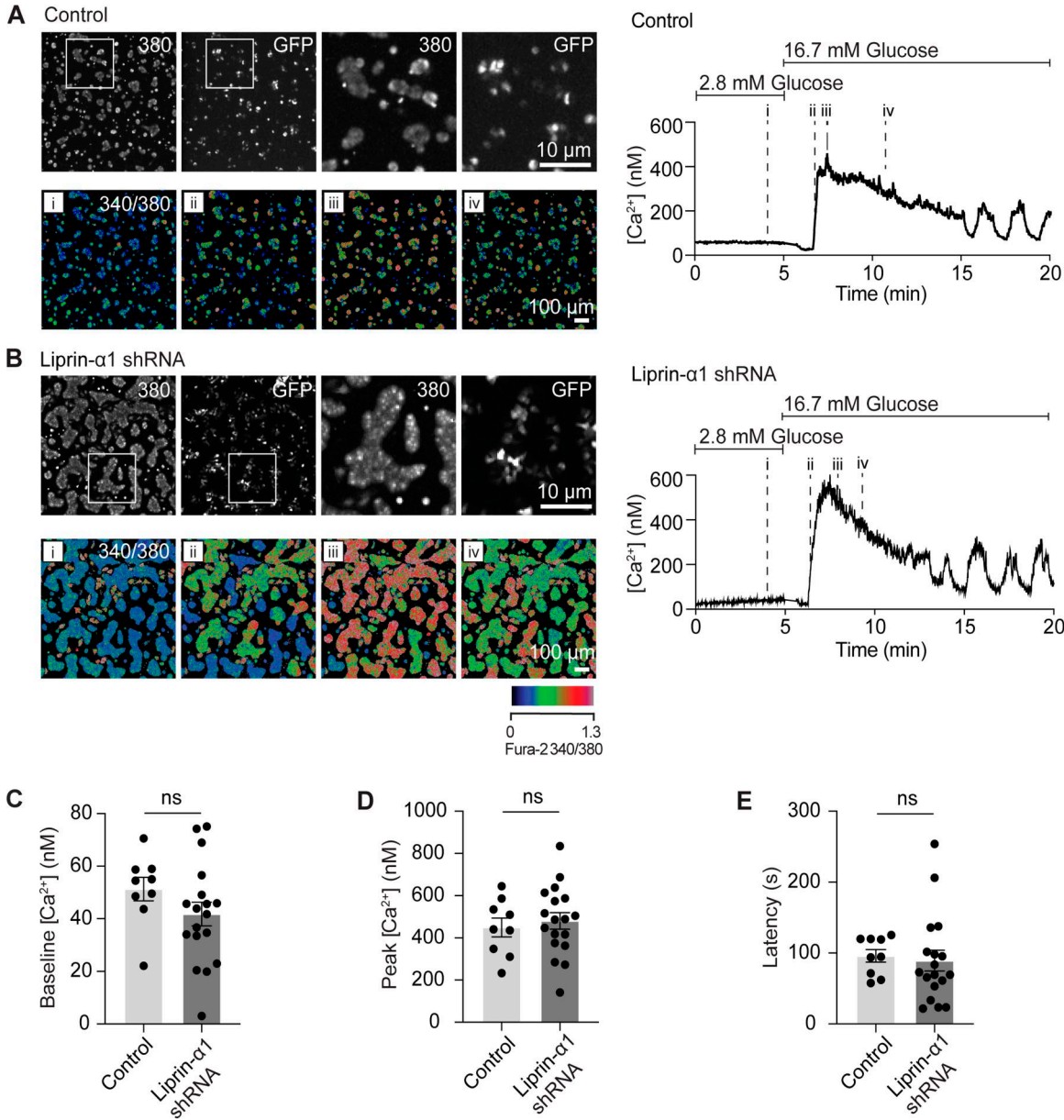

Figure S1.    **The Ca²⁺ response to glucose is not affected by liprin-α1 knock down. (A and B)** β cells (grown on laminin-511) were infected with adenovirus expressing GFP-scrambled shRNA or GFP–liprin-α1 shRNA (control). β cells were loaded with Fura-2, AM, to measure intracellular [Ca²⁺]. GFP +ve cells, indicating successful infection, were selected for analysis. Ca²⁺ responses were recorded following stimulation with 16.7 mM glucose, as shown in the pseudocolor representations of the Fura-2 340/380 ratio over four time points (i: before glucose, ii, iii and iv: after glucose). 340/380 ratios were used to calculate intracellular [Ca²⁺] according to Grynkiewicz et al. (1985), and an example Ca²⁺ response recorded within a single β cell cluster is plotted for each group, showing a robust initial rise in [Ca²⁺] followed by sustained oscillations. **(C–E)** No differences were observed in baseline [Ca²⁺] (during incubation in 2.8 mM glucose), peak [Ca²⁺], or latency (time to peak) (n ≥ 9 β cell clusters from 3 animals; Student's *t* test, unpaired, equal variance).

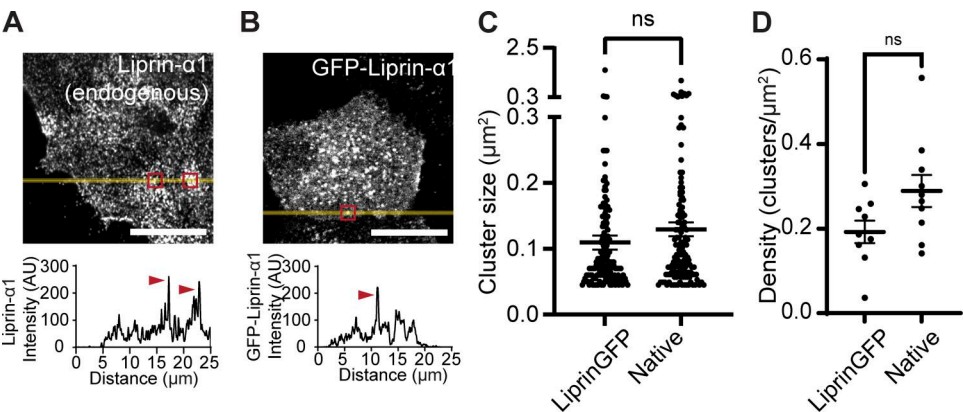

Figure S2. **GFP-liprin-α1 clusters are the same size and density as endogenous liprin-α1 clusters. (A)** Immunofluorescence staining of endogenous liprin-α1 in isolated β cells (grown on laminin-511). Liprin-α1 forms punctate spots/clusters across the β cell/laminin interface, as seen in the line scan (yellow) across a single β cell. **(B)** Live-cell super-resolution microscopy in β cells expressing GFP–liprin-α1 revealed similar punctate GFP–liprin-α1 spots/clusters, seen in the line scan (yellow) across a single β cell. **(C and D)** Characterization of GFP–liprin-α1 and native liprin-α1–immunostained cluster size (Student's t test P = 0.18, >139 clusters, 9 cells, 3 animals) and density across (Student's t test P = 0.06, 8–9 cells in each condition, n = 3 animals) the β cell–ECM interface.

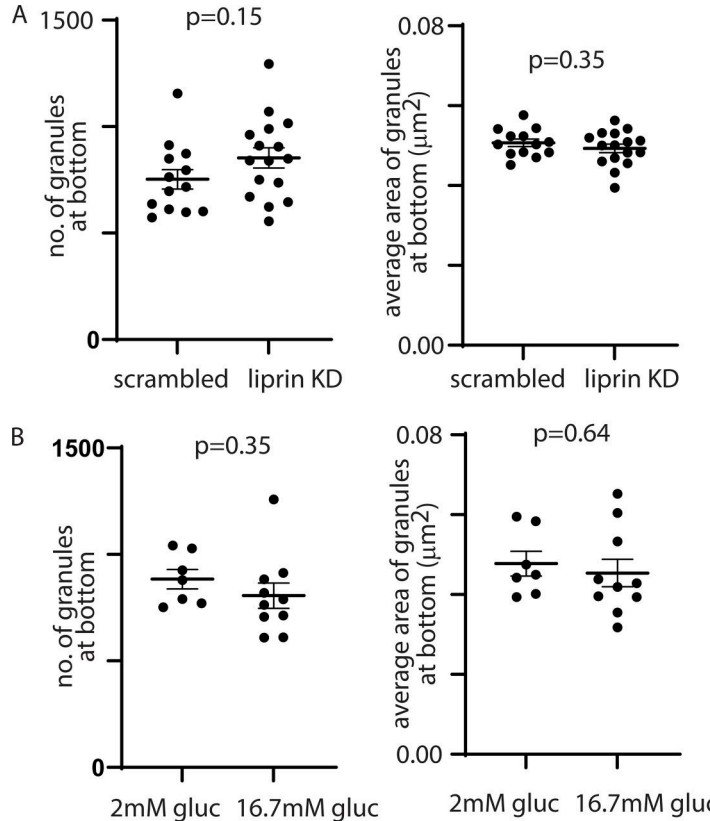

Figure S3. **Super-resolution of granules at coverslip interface show no difference in granule number or granule area with liprin-α1 knockdown or with glucose stimulation. (A)** STED Immunofluorescence images of insulin granules at the β cell–ECM interface resolved individual granules. Counts across a 25 × 25 μm area showed >1,000 granules in control cells and similar numbers (n = 3 animals, Student's t test, P = 0.15) and similar average area (n = 3 animals, Student's t test, P = 0.35) in liprin-α1 knockdown cells. **(B)** Using an identical approach and analysis to that in A counts across a 25 × 25 μm area showed >1,000 granules in control cells and similar numbers (n = 3 animals, Student's t test, P = 0.35) and similar average area (n = 3 animals, Student's t test, P = 0.64) in cells after 15 min stimulation with 16.7 mM glucose.

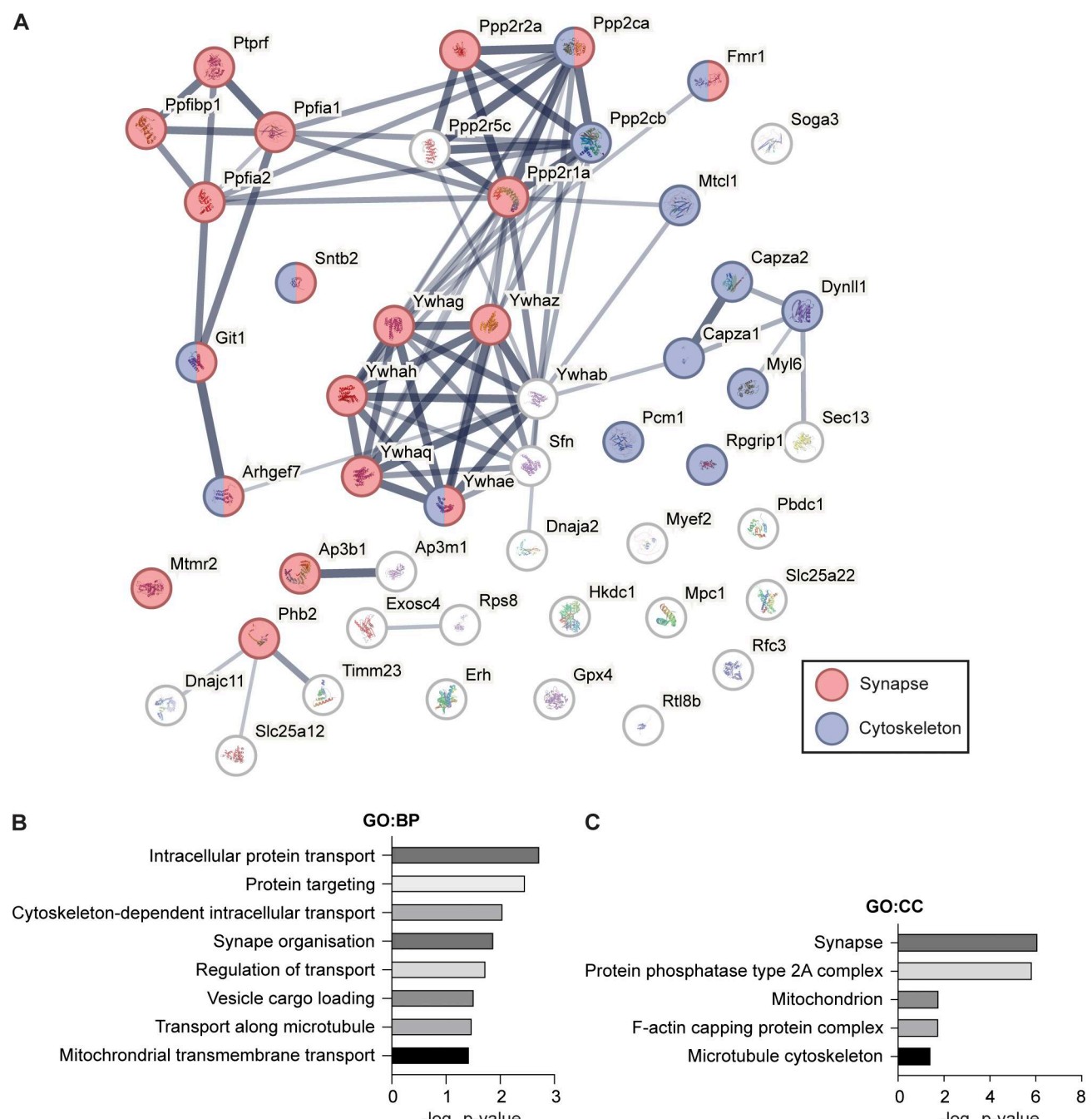

Figure S4.   **STRING analysis of proteins interacting with liprin-α1. (A)** STRING protein–protein interaction network of the 49 liprin-α1–interacting proteins using a medium (0.4) confidence level. Gene ontology (GO)-cellular component (CC) term enrichment analysis of proteins shows a significant enrichment in proteins associated with the synapse (red) and cytoskeleton (blue). **(B and C)** GO:biological process (BP) and (C) GO:CC analyses of identified liprin-α1–interacting proteins.

**Provided online is Data S1. Data S1 shows (IP-MS SAINT).**

