## [Peer Review File · The Journal of Cell Biology]

Submembrane liprin- α 1 clusters spatially localise insulin granule fusion

Kylie Jevon, kitty Sun, Nicole Hallahan, Wan Gan, Michelle Cieleish, Baharak Mahyad, Melkam Kebede, Mark Larance, and Peter Thorn

Corresponding Author(s): Peter Thorn, The University of Sydney

Review Timeline:

Submission Date:	2024-11-01
Editorial Decision:	2024-12-11
Revision Received:	2025-05-19
Editorial Decision:	2025-07-06
Revision Received:	2025-07-07

Monitoring Editor: Hongyuan Yang

Scientific Editor: Andrea Marat

Transaction Report:

DOI: <https://doi.org/10.1083/jcb.202410210>

December 11, 2024

Re: JCB manuscript #202410210

Kylie Deng
University of Sydney

Dear Ms. Deng,

Thank you for submitting your manuscript entitled "Submembrane liprin- α 1 clusters spatially localise insulin granule fusion". The manuscript was assessed by expert reviewers, whose comments are appended to this letter. We invite you to submit a revision if you can address the reviewers' key concerns, as outlined here.

You will see that all reviewers appreciated the novel insight into polarized granule targeting at the basal surface, via liprin- α 1. All reviewers sought additional details to better support the main findings. In particular, Reviewers 2 and 3 requested stronger evidence for tethering specifically by liprin- α 1, and Reviewer 1 sought demonstration that Liprin-Syntrophin interaction is required for insulin release. Reviewers also requested quantifications of some observations and greater details on methods. If possible, improving 3D imaging would be very helpful as suggested by Reviewer 3. Identifying new liprin- α 1 interaction partners, as sought by Reviewer 1, is not required in a revised manuscript.

GENERAL GUIDELINES:

Text limits: Character count for an Article is < 40,000, not including spaces. Count includes title page, abstract, introduction, results, discussion, and acknowledgments. Count does not include materials and methods, figure legends, references, tables, or supplemental legends.

Figures: Articles may have up to 10 main text figures. Figures must be prepared according to the policies outlined in our Instructions to Authors, under Data Presentation, <https://jcb.rupress.org/site/misc/ifora.xhtml>. All figures in accepted manuscripts will be screened prior to publication.

Supplemental information: There are strict limits on the allowable amount of supplemental data. Articles may have up to 5 supplemental figures. Up to 10 supplemental videos or flash animations are allowed. A summary of all supplemental material should appear at the end of the Materials and methods section.

Please note that JCB now requires authors to submit Source Data used to generate figures containing gels and Western blots with all revised manuscripts. This Source Data consists of fully uncropped and unprocessed images for each gel/blot displayed in the main and supplemental figures. Since your paper includes cropped gel and/or blot images, please be sure to provide one Source Data file for each figure that contains gels and/or blots along with your revised manuscript files. File names for Source Data figures should be alphanumeric without any spaces or special characters (i.e., SourceDataF#, where F# refers to the associated main figure number or SourceDataFS# for those associated with Supplementary figures). The lanes of the gels/blots should be labeled as they are in the associated figure, the place where cropping was applied should be marked (with a box), and molecular weight/size standards should be labeled wherever possible.

The typical timeframe for revisions is three to four months. If you anticipate any difficulties in meeting this aforementioned revision time limit, please contact us and we can work with you to find an appropriate time frame for resubmission. Please note that papers are generally considered through only one revision cycle, so any revised manuscript will likely be either accepted or rejected.

When submitting the revision, please include a cover letter addressing the reviewers' comments point by point. Please also

highlight all changes in the text of the manuscript.

Thank you for this interesting contribution to Journal of Cell Biology. You can contact us at the journal office with any questions at cellbio@rockefeller.edu.

Sincerely,

Hongyuan Yang
Monitoring Editor
Journal of Cell Biology

Tim Fessenden
Scientific Editor
Journal of Cell Biology

Reviewer #1 (Comments to the Authors (Required)):

The authors propose that insulin granule fusion in pancreatic β cells is localized to the ECM interface through the action of the scaffold protein liprin- α 1. This protein forms glucose-responsive clusters that organize granules before docking and fusion. The study employs advanced imaging, statistical analyses, and biochemical experiments to demonstrate that liprin- α 1 knockdown disrupts granule positioning and glucose-stimulated insulin secretion, with β 2-syntrophin identified as a key interacting partner. Overall, the study presents significant findings on the role of liprin- α 1 in insulin granule localization, but addressing the comments below would enhance the robustness and clarity of the manuscript.

Major Comments

1. The authors describe spatial bias in granule fusion events relative to liprin- α 1 clusters in Figures 1F, 5C, and 5D, but the quantitative approach remains vaguely described. It is unclear how distances were measured, categorized (e.g., contact vs. non-contact), or statistically compared. I recommend that the proximity threshold be defined in a dedicated methods section, along with an explanation of its biological relevance. This would provide a clearer understanding of how the threshold was chosen and strengthen the interpretation of spatial localization data.
2. Including more representative images showing how granule localization is disrupted post-liprin- α 1 knockdown, especially under glucose stimulation, would help readers better visualize the effects of liprin- α 1 loss on granule positioning (Figure 1I).
3. While the study presents evidence for the role of liprin- α 1 in granule localization, the exact molecular mechanisms remain unclear. Further experiments could help elucidate how liprin- α 1 interacts with other proteins to position insulin granules. I suggest conducting experiments to identify specific interactions between liprin- α 1 and other proteins involved in granule positioning, which would help clarify its role in the broader protein network responsible for this process.
4. The interaction between liprin- α 1 and β 2-syntrophin is an important finding, but the authors do not show how these interactions specifically contribute to insulin granule fusion. I suggest providing a detailed map of the interaction between liprin- α 1 and β 2-syntrophin and testing the impact of β 2-syntrophin knockdown on granule localization or insulin secretion. This would provide further validation of its functional role and strengthen the conclusions about how these proteins regulate granule positioning.
5. The interactome analysis in Figure 6 provides compelling evidence of liprin- α 1's interaction with β 2-syntrophin, but the strength or specificity of these interactions is not statistically quantified. Additionally, to strengthen this evidence, I recommend adding quantitative co-localization analysis, such as Pearson's or Manders' coefficients, to demonstrate how well β 2-syntrophin clusters with liprin- α 1.
6. The potential redundancy with other liprin isoforms or scaffold proteins is not addressed in the paper. Investigating whether other isoforms can compensate for liprin- α 1 knockdown would strengthen the conclusions about liprin- α 1's unique role. I recommend exploring the roles of other liprin isoforms in β cells to determine whether they have overlapping or distinct functions in granule positioning. This could provide valuable insights into the broader regulatory mechanisms in β cells.

Minor Comments

1. Ensure the labels refer to Liprin-FL and Liprin-N to match the fluorescence images and intensity chart (Figure 3D).
2. Including time-lapse images of liprin- α 1 cluster formation alongside quantitative data would enhance the manuscript by visually demonstrating how glucose stimulation affects cluster dynamics over time. This would help readers better understand the temporal changes in liprin- α 1 clustering (Figure 4).
3. Adding a panel that illustrates the spatial measurement methodology (e.g., how distances from the ECM or cluster centroids were calculated) would improve transparency in the quantification of granule proximity to liprin- α 1 clusters (Figure 5).

4. Ensure that the color of labels and figure legends match the correct colors (e.g., the notation of "laminin; red" in the figure legends for Figure 6E, but laminin is labeled in blue in the figure).
5. The manuscript mentions that liprin- α 1 knockdown reduces glucose-stimulated insulin secretion but does not affect K⁺-stimulated secretion. However, the underlying reasons for this selective effect are not fully explained. Expanding on why liprin- α 1 selectively affects glucose-stimulated secretion would clarify its specific role in granule positioning versus other stages of secretion. This should be discussed in more detail.
6. The manuscript should address potential limitations in generalizing the findings across species or different islet environments in the discussion section. Liprin- α 1 function may vary depending on the model used, and addressing this variability would provide a more balanced interpretation of the results.

Reviewer #2 (Comments to the Authors (Required)):

I agree that the paper by Deng et al is an important advance for our understanding of dense core granule exocytosis in tissue context, and that it provides novel insight into liprin-a1 function that may be relevant also in neuronal synapses. The glucose effect on liprin distribution is quite exciting for beta-cell biology. I have outlined some concerns that may help improve the paper:

The glucose dependent effect on liprin-a1 translocation to the ECM-interface and clustering is exciting, and may be related to reports that granule docking is stimulated by glucose with a similar timecourse (Straub Diabetes 2004 and Gandasi Cell Met 2018). However, my impression from Fig 4E-F is that most of the clusters simply get brighter (presumably by recruiting more liprin-a1, rather than an increase in cluster density. This is also the case for the example in 4H. Much of the increase in density in 4J may be due to that bright clusters are easier to detect. Isn't it a bit surprising then that overexpression of liprina1-N didn't increase exocytosis?

The conclusion that liprin-a1 is critical for tethering (rather than docking or priming) is perhaps consistent with the (greater) effect of liprin-a1 kd on 2nd phase secretion (Fig 2B). However, to be conclusive would require observing granules before exocytosis, ideally as they approach the ECM facing membrane. It should be established that the number of docked granules is reduced by liprin kd, as expected if it affects an upstream step. As it is, the 3D live cell imaging ignores tethered/docked granules that do not undergo exocytosis, and the insulin staining in 1A and 3C-D is of too low resolution to quantify docked granules.

Liprin kd affects glucose, but not K⁺-induced exocytosis. This is surprising if it is true that (glucose-dependent) liprin clustering is involved. Does overexpression of liprin affect cluster density or size? Do granules dock near large/bright, but not small clusters? Or is the glucose dependent effect depressed in the KCl? The latter could be tested in high glucose medium supplemented with diazoxide to suppress glucose dependent electrical activity.

There is good evidence for clustering of SNAREs and related proteins at granules, and that this is important for secretory granule docking and exocytosis. I am missing a discussion how this may relate to the similar clustering of liprin. If liprin's unique role is to direct the docking process to the ECM face, why does docking (and presumably tethering) proceed elsewhere when it is knocked down? What prevents granules from reaching areas that lack liprin?

Culture in high glucose (standard DMEM and RPMI are >10mM) stimulates exocytosis and depletes insulin granules at the membrane (Olofsson Pflugers Arch 2002), which may affect the immunostaining experiments.

3D live cell imaging: Were there any corrections applied to account for the fact that, due to geometry, exocytosis should be easier to detect in the coverslip-facing side of the cell? Could short events have been missed?

Fig 3D, individual insulin granules cannot be discerned (unlike in 3C)

Are the cells grown on laminin polarized, and does liprin-a1 kd affect cell polarity?

The introduction states that Direct measurements show that both syntaxin 1A and CaV1.2 distribute uniformly... There is good evidence for clustering of these and related proteins at granules, and that this is important for secretory granule docking and exocytosis. I am also missing a discussion how this may relate to the similar clustering of liprin.

Results: We conclude that liprin- α 1 regulates a mechanism that localises granules to the β cell/ECM-interface prior to docking. This statement lacks statistical support.

results: ...we also observed significant enrichment at the β cell/capillary-interface (labelled with laminin) like liprin- α 1, apparent when comparing the relative fluorescence intensities at the vascular, lateral and apical regions of the cells (Fig. 6 E).

What does this mean?

GFP-liprin-FL and N are not defined

Fig S2. How was cluster size calculated? Is this area or diameter?

Fig 1H and elsewhere. It is unclear if the secretion assay was done with islets, dispersed islet cells, or Min6

The 3D 2-photon imaging is not well described. What was the frame rate, and the time to switch between image planes?

In the main text, it is not always clear what sample was used.

Reviewer #3 (Comments to the Authors (Required)):

Review of Deng et al. JCB

This is an important and significant study, which falls in line with prior studies of this group and others. Mechanisms of polarized insulin secretion into the bloodstream are extremely important for physiology yet severely understudied. This study provides a substantial step forward in the field. The finding that, unlike other molecular players at these sites, liprin is dramatically recruited to the ECM contact upon glucose stimulation is striking and indicates that this protein is the major player in the process. There are several relatively small revisions, including certain controls and additional testing that would be needed prior to publication.

1. Liprin shRNA results in a very slight decrease of liprin content Fig 1G in islets.

It would be surprising if such a small reduction caused the dramatic effects described here. It is possible that the depletion is efficient but concentrated in a subpopulation of cells, especially because these appear to be whole islets. What % of cells express GFP marker after adenoviral transduction? It would be best to show liprin immunostaining in addition to WB and quantify the correlation of GFP signal to liprin intensity per cell.

2. The result of a more even secretion distribution over the cell surface in KCL is striking. The logical interpretation provided by the authors is that the secretion of already docked/available granules is not affected by liprin depletion. This result also infers that IG docking is distributed predominantly toward the ECM-associated plasma membrane and that w/o liprin IGs would be distributed evenly all over the plasma membrane. This is important, and an assumption is not enough. I do not see this directly addressed. It is recommended to analyze IG distribution at the plasma membrane at the bottom versus upper levels (similar to what is shown for rescues in Fig. 3; this would also serve as a control for Fig. 3)

In addition, the KCL result suggests that there is not only a decrease of IGs available for immediate secretion at the ECM, but also an increase of those in other regions. Does it mean that liprin is not really needed for IG recruitment to PM, only for biasing it toward the vasculature? Please discuss.

3. The authors state that "The frequency of granule fusion events in the liprin- α 1 knockdown cells was too low to use the 3D live-cell assay to map fusion events." However, it appears from the perfusion assays that the secretion decrease, especially in the first phase, is less than 50%. Thus, it should be feasible to test whether localization of those events occurs evenly all over the membrane, like in KCL, or not. This would show the liprin-dependent effect of secretion of glucose-dependent newly delivered/docked granules. Another way to look at this distribution would be to stimulate with glucose plus KCl combined, which is known to induce a significant secretion yet would show the events dependent on IG delivery.

4. The data in Fig 4 showing liprin dynamics in high glucose are very significant. They show the unique sensitivity of this protein to glucose signaling and indicate that this is likely a critical regulator of directed IG recruitment for secretion. However, a question arises: Does IG concentration at the cell bottom (docking) correlate with liprin intensity over time? This is an important point for this study and needs to be addressed.

5. Lack of liprin interaction with RIMs: this is a surprising negative result. co-IP is, of course, often challenging and might not be the most sensitive test for protein interactions. It would be important to test this interaction further, possibly using the C-terminus of liprin instead of the FL protein, and/or using a cross-linking co-IP approach, or a proximity ligation approach. Also, it is possible that this interaction is detectable in primary beta cells while it is not detected in Min6. Min6 is a nice model for in vitro studies but there are many differences from primary cells. Would it be possible to test this interaction in islets?

Minor point: please make sure that all abbreviations are spelled out before: e.g. full length (FL)

RESPONSE TO REFEREES

We thank the referees for their time and comments. In response we have conducted more experiments, conducted more analysis and revised the figures and text. We believe this has improved the manuscript and hope it is now acceptable for publication.

Reviewer #1 (Comments to the Authors (Required)):

The authors propose that insulin granule fusion in pancreatic β cells is localized to the ECM interface through the action of the scaffold protein liprin- α 1. This protein forms glucose-responsive clusters that organize granules before docking and fusion. The study employs advanced imaging, statistical analyses, and biochemical experiments to demonstrate that liprin- α 1 knockdown disrupts granule positioning and glucose-stimulated insulin secretion, with β 2-syntrophin identified as a key interacting partner. Overall, the study presents significant findings on the role of liprin- α 1 in insulin granule localization, but addressing the comments below would enhance the robustness and clarity of the manuscript.

Major Comments

1. The authors describe spatial bias in granule fusion events relative to liprin- α 1 clusters in Figures 1F, 5C, and 5D, but the quantitative approach remains vaguely described. It is unclear how distances were measured, categorized (e.g., contact vs. non-contact), or statistically compared. I recommend that the proximity threshold be defined in a dedicated methods section, along with an explanation of its biological relevance. This would provide a clearer understanding of how the threshold was chosen and strengthen the interpretation of spatial localization data.

We thank the referee for highlighting explanations we overlooked. For the 3D mapping of exocytosis we use a simple definition of contact that does not require thresholding. When focussed at the coverslip, the first optical 2-photon section detects granule fusion events within 1 μ m of the coverslip surface (which we identify as "bottom" and the second and subsequent optical sections detect only events that are >1 μ m away. For granule fusion proximity analyses (Fig 5), our original measurements were taken manually by drawing lines from the point of highest SRB signal intensity to its nearest GFP-liprin neighbours and recording the shortest distance. In response to the referee we have now revised this analysis using a more reproducible method involving automated distance measurements using the DiAna plugin on ImageJ. We have amended the methods to include a paragraph detailing the parameters we used to perform the GFP-liprin segmentation, distance and contact/non-contact analysis. For liprin contact vs. non-contact analysis, we performed object-based colocalisation analysis using segmented GFP-liprin and circular ROIs (diameter 0.3 μ m; the average size of an insulin granule) centred over the point of highest SRB signal intensity. Granule fusion events were classified as 'contacting liprin' if there was any amount of overlap between the segmented granule fusion ROI and GFP-liprin, and 'non-contacting' if there was no overlap.

2. Including more representative images showing how granule localization is disrupted post-liprin- α 1 knockdown, especially under glucose stimulation, would help readers better visualize the effects of liprin- α 1 loss on granule positioning (Figure 1I).

In response we have developed a new way of visualising the data that complements the previous exemplar 3D projections and histograms. The exocytic density in each optical section is shown as dots, with the number of dots directly proportionate to the mean density

within each section, across the whole dataset (see new Fig 1J, K, L). This representation complements the exemplar images and more clearly shows the impact of liprin- α 1 knockdown on granule fusion localisation and the partial but significant recovery with the rescue experiment.

3. While the study presents evidence for the role of liprin- α 1 in granule localization, the exact molecular mechanisms remain unclear. Further experiments could help elucidate how liprin- α 1 interacts with other proteins to position insulin granules. I suggest conducting experiments to identify specific interactions between liprin- α 1 and other proteins involved in granule positioning, which would help clarify its role in the broader protein network responsible for this process.

We do agree with the referee that this now becomes an important question. However, our work is already extensive and establishes the foundational evidence that a presynaptic-like complex regulates the positioning of granule fusion.

4. The interaction between liprin- α 1 and β 2-syntrophin is an important finding, but the authors do not show how these interactions specifically contribute to insulin granule fusion. I suggest providing a detailed map of the interaction between liprin- α 1 and β 2-syntrophin and testing the impact of β 2-syntrophin knockdown on granule localization or insulin secretion. This would provide further validation of its functional role and strengthen the conclusions about how these proteins regulate granule positioning.

Linking liprin- α 1 to β 2-syntrophin by showing they colocalize (correlation coefficients now shown in this revision) and coimmunoprecipitate is an exciting finding in our paper. β 2-syntrophin is already known to be present in β cells and reduced expression of β 2-syntrophin affects granule mobility and insulin secretion (Schubert et al 2010), both findings consistent with our new work and with an interaction with liprin- α 1. It goes beyond this paper, but we agree with the referee identifying the regions of interaction between liprin and β 2-syntrophin are important next steps.

5. The interactome analysis in Figure 6 provides compelling evidence of liprin- α 1's interaction with β 2-syntrophin, but the strength or specificity of these interactions is not statistically quantified. Additionally, to strengthen this evidence, I recommend adding quantitative co-localization analysis, such as Pearson's or Manders' coefficients, to demonstrate how well β 2-syntrophin clusters with liprin- α 1.

We thank the referee for this suggestion and we have performed a Pearson's colocalisation analysis which shows significance, see new Fig 6E. In addition the IP-MS work was statistically analysed using the SAINT approach which uses a Bayesian False Discovery Rate (BFDR) as a statistical identifier of significance (see Fig 6b, S4 and Dataset S1) which, in response to the referee we have now highlighted in the text. Prey proteins were considered significant if they passed the Bayesian FDR cutoff of <0.05. BFDR for each significant prey protein are provided in Supplementary Dataset S1.

6. The potential redundancy with other liprin isoforms or scaffold proteins is not addressed in the paper. Investigating whether other isoforms can compensate for liprin- α 1 knockdown would strengthen the conclusions about liprin- α 1's unique role. I recommend exploring the roles of other liprin isoforms in β cells to determine whether they have overlapping or distinct functions in granule positioning. This could provide valuable insights into the broader regulatory mechanisms in β cells.

The referee is correct, the role of the broader liprin family in regulating insulin secretion now becomes an interesting question. Our analysis positively identifies liprin- α 2 and liprin- β 1 as binding partners for liprin- α 1. Both are exciting candidates to pursue, for example there is

good evidence from neurones that liprin- α 2 interacts with ELKS – that would be important to test in β cells. Less is known about liprin- β 1 but evidence in epithelial cells shows an interaction with the scaffold protein KANK which in turn binds with the focal adhesion protein talin. In the context of β cells, a liprin- β 1_KANK_talin complex would be evidence for a long-sought mechanism that locates the presynaptic complex to the sites of integrin activation at the capillary interface. So, there is a lot to uncover. In this paper we chose to first study liprin- α 1 because of its relative abundance, we are in fact pursuing liprin- β 1 and have evidence to support a KANK1 interaction but to progress the work further, along the lines the referee suggests, we believe is beyond the scope of the current work.

Minor Comments

1. Ensure the labels refer to Liprin-FL and Liprin-N to match the fluorescence images and intensity chart (Figure 3D).

Thank you, DONE

2. Including time-lapse images of liprin- α 1 cluster formation alongside quantitative data would enhance the manuscript by visually demonstrating how glucose stimulation affects cluster dynamics over time. This would help readers better understand the temporal changes in liprin- α 1 clustering (Figure 4).

The kymograph images show these changes already and are an excellent representation of both the size and temporal dynamics of liprin- α 1 clusters (Fig. 4F) as well as their association with the time and location of insulin granule fusion events.

3. Adding a panel that illustrates the spatial measurement methodology (e.g., how distances from the ECM or cluster centroids were calculated) would improve transparency in the quantification of granule proximity to liprin- α 1 clusters (Figure 5).

This is a good point and in response we have now included details in the fig legend and methods.

4. Ensure that the color of labels and figure legends match the correct colors (e.g., the notation of "laminin; red" in the figure legends for Figure 6E, but laminin is labeled in blue in the figure).

Thank you, DONE.

5. The manuscript mentions that liprin- α 1 knockdown reduces glucose-stimulated insulin secretion but does not affect K⁺-stimulated secretion. However, the underlying reasons for this selective effect are not fully explained. Expanding on why liprin- α 1 selectively affects glucose-stimulated secretion would clarify its specific role in granule positioning versus other stages of secretion. This should be discussed in more detail.

We have expended on this in the discussion and agree it is a very interesting finding. We now suggest and discuss the idea that that liprin- α 1 is involved in the “amplifying” pathway of glucose regulated secretion.

6. The manuscript should address potential limitations in generalizing the findings across species or different islet environments in the discussion section. Liprin- α 1 function may vary depending on the model used, and addressing this variability would provide a more balanced interpretation of the results.

We agree and have added a paragraph in the discussion. We know liprin- α 1 is present in human β cells and is enriched at the capillary interface which suggests it may have a similar

function in human compared with mouse. However, we would expect that different isoforms and levels of control might exist in human and we have discussed that point.

Reviewer #2 (Comments to the Authors (Required)):

I agree that the paper by Deng et al is an important advance for our understanding of dense core granule exocytosis in tissue context, and that it provides novel insight into liprin-a1 function that may be relevant also in neuronal synapses. The glucose effect on liprin distribution is quite exciting for beta-cell biology. I have outlined some concerns that may help improve the paper:

The glucose dependent effect on liprin-a1 translocation to the ECM-interface and clustering is exciting, and may be related to reports that granule docking is stimulated by glucose with a similar timecourse (Straub Diabetes 2004 and Gandasi Cell Met 2018). However, my impression from Fig 4E-F is that most of the clusters simply get brighter (presumably by recruiting more liprin-a1, rather than an increase in cluster density. This is also the case for the example in 4H. Much of the increase in density in 4J may be due to that bright clusters are easier to detect. Isn't it a bit surprising then that overexpression of liprina1-N didn't increase exocytosis?

Both size and number of clusters increase (Fig 4K,L) which is very strong evidence for a translocation of liprin- α 1. Whether there is a bias, as the referee suggests, for recruitment to an already present cluster, is not clear. Liprin- α can undergo phase separation and that has been linked with function. It is unclear if that is what we are seeing and further work will be needed.

Our interpretation of the lack of increase in exocytosis with liprin-full length or liprin-N overexpression would be that there is another limiting step in the processes. Given the complexity of docking, priming and fusion it would not be surprising if any one of these steps was rate-limiting.

The conclusion that liprin-a1 is critical for tethering (rather than docking or priming) is perhaps consistent with the (greater) effect of liprin-a1 kd on 2nd phase secretion (Fig 2B). However, to be conclusive would require observing granules before exocytosis, ideally as they approach the ECM facing membrane. It should be established that the number of docked granules is reduced by liprin KD, as expected if it affects an upstream step. As it is, the 3D live cell imaging ignores tethered/docked granules that do not undergo exocytosis, and the insulin staining in 1A and 3C-D is of too low resolution to quantify docked granules.

The referee raises interesting points and in response we have turned to super-resolution imaging (which readily resolves single granules) of submembrane insulin granules under conditions of stimulation and liprin KD. These images reveal 1000s of granules at the cell-coverslip interface with analysis showing no differences in granule number whether the cells are stimulated or not and no differences with liprin KD. This data indicates that although liprin might be tethering the granules (we are unsure of the relationship with granule docking) it does not impact on the overall number of granules close to the membrane. At least in part this is probably a reflection of the very low proportion of granules, compared to the total pool, that undergo exocytosis.

Liprin kd affects glucose, but not K⁺-induced exocytosis. This is surprising if it is true that

(glucose-dependent) liprin clustering is involved. Does overexpression of liprin affect cluster density or size? Do granules dock near large/bright, but not small clusters? Or is the glucose dependent effect depressed in the KCI? The latter could be tested in high glucose medium supplemented with diazoxide to suppress glucose dependent electrical activity. *We cannot rule out that expression of liprin GFP might affect clustering. However, Fig 4A-D shows immunostaining of native liprin- α 1 and that shows clustering and Fig S2 shows the similarity of side-by-side comparison of native and GFP clusters. We have not segregated cluster brightness or mixed stimuli but our key finding of glucose dependence of liprin- α 1 is exciting and deserves further study.*

There is good evidence for clustering of SNAREs and related proteins at granules, and that this is important for secretory granule docking and exocytosis. I am missing a discussion how this may relate to the similar clustering of liprin. If liprin's unique role is to direct the docking process to the ECM face, why does docking (and presumably tethering) proceed elsewhere when it is knocked down? What prevents granules from reaching areas that lack liprin?

Yes this is an important issue, which we address with a revised discussion.

Culture in high glucose (standard DMEM and RPMI are >10mM) stimulates exocytosis and depletes insulin granules at the membrane (Olofsson Pflugers Arch 2002), which may affect the immunostaining experiments.

The referee is correct the cells are cultured under conditions of steady stimulation. For our immunostaining however comparisons of unstimulated vs stimulated (eg Fig. 4) we decrease the glucose to 2.8 mM for 30 minutes prior to stimulation which (in secretion assays) resets the cells to basal secretion.

3D live cell imaging: Were there any corrections applied to account for the fact that, due to geometry, exocytosis should be easier to detect in the coverslip-facing side of the cell? Could short events have been missed?

Geometry and sample rates might mean we miss some granule fusion events. It is however unlikely that detection is easier at the coverslip, the cells are no more than 10 μ m thick and this is well within the depth parameters for our multiphoton microscope, the events are the same brightness and size wherever they occur around the cell. We have directly previously tested the speed issue; granule fusion events have lifetimes of 10s of seconds and even when we speed up acquisition, we do not see more events (see Low et al 2013 for details when we first used this approach).

Fig 3D, individual insulin granules cannot be discerned (unlike in 3C) Are the cells grown on laminin polarized, and does liprin- α 1 kd affect cell polarity? *Our previous work directly shows that cells grown on laminin substrates do polarise as demonstrated by the distribution of polar determinants (Jevon et al 2022). In this paper we show liprin- α 1 polarisation with enrichment at the coverslip β cell interface (Fig. 1A) and loss of polarity with the liprin- α 1 N-terminus mutant.*

The introduction states that Direct measurements show that both syntaxin 1A and CaV1.2 distribute uniformly... There is good evidence for clustering of these and related proteins at granules, and that this is important for secretory granule docking and exocytosis. I am also missing a discussion how this may relate to the similar clustering of liprin. *Yes, this is a good point, and we have revised the introduction to consider structural clustering (for syntaxin) and functional local activation (for CaV1.2). Our data show that exocytosis is spatially coupled with liprin- α 1 clusters (Fig 5) and although liprin- α 1 IP did not*

pull down either syntaxin or Cav we predict all these proteins are likely components of a large macromolecular presynaptic-like complex. However, the precision of glucose-dependent tethering, docking, priming and fusion of granules strongly suggest an intricate control of the spatial and temporal organisation of this complex. In response to the referee's comments, we have expanded the discussion to include speculation as to how the dynamic liprin- α 1 clustering might be related to the more distal elements that control docking and priming.

Results: We conclude that liprin- α 1 regulates a mechanism that localises granules to the β cell/ECM-interface prior to docking. This statement lacks statistical support. *The referee is correct, without direct evidence conclude is too strong, we have changed it to "suggest".*

results: ...we also observed significant enrichment at the β cell/capillary-interface (labelled with laminin) like liprin- α 1, apparent when comparing the relative fluorescence intensities at the vascular, lateral and apical regions of the cells (Fig. 6 E). What does this mean?

We have added more detail to clarify this statement. This is a reference to β cell polarisation where the capillary interface is defined as basal, cell-cell contacts lateral and the abvascular region as apical – each of these domains is defined by the presence of polarity determinant proteins and we have added the reference to this work (Gan et al 2017). In the context of this current paper liprin- α 1 and β 2-syntrophin both show enrichment in the basal region where the β cells contact the capillaries.

GFP-liprin-FL and N are not defined

DONE

Fig S2. How was cluster size calculated? Is this area or diameter?

Cluster size is area and we have now included revisions to this section to explain how we did the analysis, particularly with respect to the spatial analysis of granule fusion with liprin clusters.

Fig 1H and elsewhere. It is unclear if the secretion assay was done with islets, dispersed islet cells, or Min6

All of the secretion assays have been done with native mouse cells. We have added wording though the manuscript to reinforce this. The only MIN6 work was the IP/mass spec.

The 3D 2-photon imaging is not well described. What was the frame rate, and the time to switch between image planes?

We thank the referee for highlighting this and a description is now included in the methods.

In the main text, it is not always clear what sample was used.

We have checked this and revised where necessary.

Reviewer #3 (Comments to the Authors (Required)):

Review of Deng et al. JCB

This is an important and significant study, which falls in line with prior studies of this group and others. Mechanisms of polarized insulin secretion into the bloodstream are extremely important for physiology yet severely understudied. This study provides a substantial step forward in the field. The finding that, unlike other molecular players at these sites, liprin is dramatically recruited to the ECM contact upon glucose stimulation is striking and indicates that this protein is the major player in the process. There are several relatively small revisions, including certain controls and additional testing that would be needed prior to publication.

1. Liprin shRNA results in a very slight decrease of liprin content Fig 1G in islets. It would be surprising if such a small reduction caused the dramatic effects described here. It is possible that the depletion is efficient but concentrated in a subpopulation of cells, especially because these appear to be whole islets. What % of cells express GFP marker after adenoviral transduction? It would be best to show liprin immunostaining in addition to WB and quantify the correlation of GFP signal to liprin intensity per cell.

The decrease we see is significant and although it is difficult to define an absolute threshold imaging of GFP expression shows that almost all cells are infected. The robust decrease and then rescue of secretion shows a very strong impact of our manipulation of liprin expression. Furthermore, the GFP tag enables us to positively identify the infected cells in our live-cell imaging studies.

2. The result of a more even secretion distribution over the cell surface in KCL is striking. The logical interpretation provided by the authors is that the secretion of already docked/available granules is not affected by liprin depletion. This result also infers that IG docking is distributed predominantly toward the ECM-associated plasma membrane and that w/o liprin IGs would be distributed evenly all over the plasma membrane. This is important, and an assumption is not enough. I do not see this directly addressed. It is recommended to analyze IG distribution at the plasma membrane at the bottom versus upper levels (similar to what is shown for rescues in Fig. 3; this would also serve as a control for Fig. 3)

To answer this question we performed STED super-resolution which resolves individual granules and identified all the granules at the ECM-interface. We compared with/without glucose stimulation and after liprin KD. In all cases there was no difference in granule numbers (new Fig. S3). We discuss this result in the paper and suggest that the total granule population in this region (1000s granules) is so much higher than the sub-population of granules that actually fuse (10s granules) that it is perhaps not surprising that we can detect no differences. Our evidence suggests that liprin- α 1 is not involved with docking, priming and fusion (see Figs 1-2) however, an important next step will be to determine the relationship between liprin- α 1 interactions with granules these distal steps.

In addition, the KCL result suggests that there is not only a decrease of IGs available for immediate secretion at the ECM, but also an increase of those in other regions. Does it mean that liprin is not really needed for IG recruitment to PM, only for biasing it toward the vasculature? Please discuss.

In response we have added commentary through the manuscript to allude to this. For high K stimulation, all we see is a disruption of localisation with liprin KD and so yes, for high K the referee is correct. As discussed in the paper we think this places liprin- α 1 before docking,

priming and fusion (because the number of granules fusing, and overall insulin secretion, induced by high K, is not affected). However, liprin- α 1 is probably doing more than "just" positioning granules to the ECM interface because knockdown does reduce the amount of glucose-induced insulin secretion. Because of the data showing liprin- α 1 translocation in response to glucose (Fig 4) we think that liprin- α 1 is under glucose-dependent regulation. However, we can't exclude the possibility of a separate glucose-dependent mechanism that targets granules to the ECM interface (eg microtubules) that are then tethered by liprin- α 1.

3. The authors state that "The frequency of granule fusion events in the liprin- α 1 knockdown cells was too low to use the 3D live-cell assay to map fusion events." However, it appears from the perfusion assays that the secretion decrease, especially in the first phase, is less than 50%. Thus, it should be feasible to test whether localization of those events occurs evenly all over the membrane, like in KCL, or not. This would show the liprin-dependent effect of secretion of glucose-dependent newly delivered/docked granules. Another way to look at this distribution would be to stimulate with glucose plus KCl combined, which is known to induce a significant secretion yet would show the events dependent on IG delivery. *Although our 3D assay robustly detects events the analysis fails when the number of fusion events per cell is low. We can however directly answer the referee's question of the impact of liprin distribution on glucose-induced granule positioning and show in Fig 3G, H, I that liprin mislocalisation (in this case using liprin-N overexpression) correlates with mispositioning of granule fusion induced by high glucose.*

4. The data in Fig 4 showing liprin dynamics in high glucose are very significant. They show the unique sensitivity of this protein to glucose signaling and indicate that this is likely a critical regulator of directed IG recruitment for secretion. However, a question arises: Does IG concentration at the cell bottom (docking) correlate with liprin intensity over time? This is an important point for this study and needs to be addressed.

As stated above we have addressed this issue using STED super resolution microscopy and see no difference in the granule numbers in the sub-membrane ECM interface. We think this reflects the proportionately very small number of granules that dock and fuse. As the referee will be aware, granule dynamics prior to fusion in β cells, is an interesting and controversial topic. We think that liprin- α 1 tethers granules prior to docking but this tethering is likely to be transient, stochastic and affect very small numbers of granules compared to the entire granule population.

5. Lack of liprin interaction with RIMs: this is a surprising negative result. co-IP is, of course, often challenging and might not be the most sensitive test for protein interactions. It would be important to test this interaction further, possibly using the C-terminus of liprin instead of the FL protein, and/or using a cross-linking co-IP approach, or a proximity ligation approach. Also, it is possible that this interaction is detectable in primary beta cells while it is not detected in Min6. Min6 is a nice model for in vitro studies but there are many differences from primary cells. Would it be possible to test this interaction in islets?

We agree, the limitations of methods and preparations make interpretation difficult, and this applies to both our work and the work of others which means that comparisons are problematic. It would be great to test in primary islets but we know that β cell interactions with capillaries are disrupted (Jevon et al 2022) and so isolated islets are not a good model.

The data we show with the liprin mutants are promising and in the future we plan to use these to try to define the interactome and approaches like cross-linking and PLA as the referee suggests will be important. However, we need to develop better models to really

dissect out this complex. In our hands pancreatic slices are the best system that preserves the interactions but IP with this system is not possible. We do think that the culture of cells on to laminin-coated coverslips is a valuable system – it does reproduce cell polarity, organisation and targeting of insulin granule fusion – however, glucose sensitivity is different (compared with slices) and so there are deficiencies with this approach. And, of course, what we are really interested in are human β cells and that will require better in vitro models.

Minor point: please make sure that all abbreviations are spelled out before: e.g. full length (FL)

Thank you – DONE.

July 6, 2025

RE: JCB Manuscript #202410210R

Peter Thorn
The University of Sydney

Dear Dr. Thorn:

Thank you for submitting your revised manuscript entitled "Submembrane liprin- α 1 clusters spatially localise insulin granule fusion". The reviewers now support publication so we would be happy to publish your paper in JCB pending final revisions necessary to meet our formatting guidelines (see details below).

A. MANUSCRIPT ORGANIZATION AND FORMATTING:

- 1) Text limits: Character count for Articles is < 40,000, not including spaces. Count includes abstract, introduction, results, discussion, and acknowledgments. Count does not include title page, figure legends, materials and methods, references, tables, or supplemental legends.
- 2) Figures limits: Articles may have up to 10 main text figures.
- 3) Figure formatting: Scale bars must be present on all microscopy images, including inset magnifications. Molecular weight or nucleic acid size markers must be included on all gel electrophoresis. Aspect ratios of images may not be altered.
- 4) Statistical analysis: Error bars on graphic representations of numerical data must be clearly described in the figure legend. The number of independent data points (n) represented in a graph must be indicated in the legend. Statistical methods should be explained in full in the materials and methods. For figures presenting pooled data the statistical measure should be defined in the figure legends. Please also be sure to indicate the statistical tests used in each of your experiments (either in the figure legend itself or in a separate methods section) as well as the parameters of the test (for example, if you ran a t-test, please indicate if it was one- or two-sided, etc.). Also, if you used parametric tests, please indicate if the data distribution was tested for normality (and if so, how). If not, you must state something to the effect that "Data distribution was assumed to be normal but this was not formally tested."
- 5) Abstract and title: The abstract should be no longer than 160 words and should communicate the significance of the paper for a general audience. The title should be less than 100 characters including spaces. Make the title concise but accessible to a general readership.
- 6) Materials and methods: Should be comprehensive and not simply reference a previous publication for details on how an experiment was performed. Please provide full descriptions in the text for readers who may not have access to referenced manuscripts.
- 7) All antibodies, cell lines, animals, and tools used in the manuscript should be described in full, including accession numbers for materials available in a public repository such as the Resource Identification Portal. Please be sure to provide the sequences for all of your primers/oligos and RNAi constructs in the materials and methods. You must also indicate in the methods the source, species, and catalog numbers (where appropriate) for all of your antibodies. Please also indicate the acquisition and quantification methods for immunoblotting/western blots.
- 8) Microscope image acquisition: The following information must be provided about the acquisition and processing of images:
 - a. Make and model of microscope
 - b. Type, magnification, and numerical aperture of the objective lenses
 - c. Temperature
 - d. Imaging medium
 - e. Fluorochromes
 - f. Camera make and model
 - g. Acquisition software
 - h. Any software used for image processing subsequent to data acquisition. Please include details and types of operations involved (e.g., type of deconvolution, 3D reconstitutions, surface or volume rendering, gamma adjustments, etc.).

10) Supplemental materials: There are strict limits on the allowable amount of supplemental data. Articles may have up to 5 supplemental figures. Please also note that tables, like figures, should be provided as individual, editable files. A summary of all supplemental material should appear at the end of the Materials and methods section.

13) ORCID IDs: ORCID IDs are unique identifiers allowing researchers to create a record of their various scholarly contributions in a single place. Please note that ORCID IDs are now *required* for all authors. At resubmission of your final files, please be sure to provide your ORCID ID and those of all co-authors.

Please note that JCB now requires authors to submit Source Data used to generate figures containing gels and Western blots with all revised manuscripts. This Source Data consists of fully uncropped and unprocessed images for each gel/blot displayed in the main and supplemental figures. For assays performed using capillary electrophoresis and/or immunoassay-based detection, authors should instead provide the electropherogram graph(s) for each experiment, plotting fluorescence/chemiluminescence intensity vs. molecular weight/size. Please be sure to provide one Source Data file for each figure gels, blots, and/or capillary electrophoresis assays along with your revised manuscript files. File names for Source Data figures should be alphanumeric without any spaces or special characters (i.e., SourceDataF#, where F# refers to the associated main figure number or SourceDataFS# for those associated with Supplementary figures). For traditional gels and blots, the lanes of the gels/blots should be labeled as they are in the associated figure, the place where cropping was applied should be marked (with a box), and molecular weight/size standards should be labeled wherever possible. For capillary electrophoresis assays, each trace in the graph should be color-coded and labeled to indicate which protein, gene, or sample is being measured (please try to avoid red/green combinations to accommodate our color-blind readers).

Journal of Cell Biology now requires a data availability statement for all research article submissions. These statements will be published in the article directly above the Acknowledgments. The statement should address all data underlying the research presented in the manuscript. Please visit the JCB instructions for authors for guidelines and examples of statements at (<https://rupress.org/jcb/pages/editorial-policies#data-availability-statement>).

B. FINAL FILES:

****It is JCB policy that if requested, original data images must be made available to the editors. Failure to provide original images upon request will result in unavoidable delays in publication. Please ensure that you have access to all original data images prior to final submission.****

****The license to publish form must be signed before your manuscript can be sent to production. A link to the electronic license to publish form will be sent to the corresponding author only. Please take a moment to check your funder requirements before choosing the appropriate license.****

Thank you for your attention to these final processing requirements. Please revise and format the manuscript and upload materials within 7 days. If you need an extension for whatever reason, please let us know and we can work with you to determine a suitable revision period.

Thank you for this interesting contribution, we look forward to publishing your paper in Journal of Cell Biology.

Sincerely,

Hongyuan Yang, PhD
Monitoring Editor

Andrea L. Marat, PhD
Deputy Editor

Journal of Cell Biology

Reviewer #1 (Comments to the Authors (Required)):

The authors have successfully addressed most of my comments, leading to significant improvements. Notably, the previous manual proximity measurements have been replaced with a more robust and transparent method, which yielded consistent results and enhanced data clarity. The newly added dot-density plots effectively demonstrate the impact of liprin- α 1 knockdown on granule localization. Furthermore, the authors quantified interactions between liprin- α 1 and β 2-syntrophin using co-localization metrics and supported their IP-MS findings with appropriate statistical validation. Although further mechanistic insights and functional assays regarding β 2-syntrophin are subjects for future studies, the current scope of the work is clearly defined and appropriate. Minor issues, such as figure labeling and the differentiation between glucose and K⁺ stimulation in the discussion, have been satisfactorily resolved. Overall, the revisions have improved both the clarity and scientific rigor of the manuscript.

Reviewer #2 (Comments to the Authors (Required)):

no further comments.

Reviewer #3 (Comments to the Authors (Required)):

This study is an important contribution to the field. The authors have addressed all our comments in this revised manuscript. The revised study is recommended for publication.